# Measuring Audio's Impact on Correctness: Audio-Contribution-Aware Post-Training of Large Audio Language Models

**Haolin He**[1,3,*]**, Xingjian Du**[2,*]**, Renhe Sun**[3,*]**, Zheqi Dai**[1]**, Yujia Xiao**[1]**, Mingru Yang**[3]**, Jiayi Zhou**[3]**, Xiquan Li**[4]**, Zhengxi Liu**[1]**, Zining Liang**[1]**, Chunyat Wu**[1]**, Qianhua He**[5]**, Tan Lee**[1]**, Xie Chen**[4]**, Wei-Long Zheng**[4]**, Weiqiang Wang**[3]**, Mark Plumbley**[6]**, Jian Liu**[3,†]**, Qiuqiang Kong**[1,†]

[1]The Chinese University of Hong Kong, Hong Kong, China     [2]University of Rochester, USA
[3]Ant Group, China     [4]Shanghai Jiao Tong University, China
[5]South China University of Technology, China     [6]King's College London, UK
rex.lj@antgroup.com    qqkong@ee.cuhk.edu.hk

## Abstract

Large Audio Language Models (LALMs) represent an important frontier in multimodal AI, addressing diverse audio tasks. Recently, post-training of LALMs has received increasing attention due to significant performance improvements over foundation models. While single-stage post-training such as reinforcement learning (RL) has demonstrated promising results, multi-stage approaches such as supervised fine-tuning (SFT) followed by RL remain suboptimal. The allocation of data across multiple training stages to maximize LALM capabilities has not been fully explored, and large-scale, high-quality datasets for such research are also lacking. To address these problems, we firstly present AudioMCQ, a comprehensive audio multiple-choice question dataset comprising 571k samples with two kinds of chain-of-thought annotations. Secondly, we investigate the prevalent zero audio-contribution phenomenon in LALMs, where models derive correct answers solely from textual information without processing audio content. We propose Audio-Contribution Filtering to partition data into weak and strong audio-contribution subsets. Based on these insights, we develop two effective post-training paradigms: Weak-to-Strong (SFT on weak audio-contribution data followed by RL on strong audio-contribution data) and Mixed-to-Strong (SFT on mixed audio-contribution data followed by RL on strong audio-contribution data). We achieve first place in the DCASE 2025 Audio-Question-Answering challenge by using AudioMCQ. Additionally, leveraging our dataset with different training strategies, we achieve 78.2% on MMAU-test-mini, 75.6% on MMAU, 67.0% on MMAR, and 71.7% on MMSU, establishing new state-of-the-art performance.

## 1 Introduction

In recent years, Large Audio Language Models (LALMs) have received increasing attention in the field of multimodal artificial intelligence (Yang et al., 2025c; Zhao et al., 2023). These LALMs are designed to handle diverse audio-related tasks, including Automatic Speech Recognition (ASR) (Benzeghiba et al., 2007), Audio Captioning (AC) (Xu et al., 2023), and Music Captioning (MC) (Manco et al., 2021), among others. To achieve effective audio-text alignment in the latent space while developing multi-task execution capabilities, the pre-training of LALMs requires substantial computational resources and extensive datasets. Given these resource-intensive requirements, post-training of LALMs has emerged as an promising research direction.

Current post-training research for LALMs focuses on two primary areas. First, recent work aims to enable LALMs to perform chain-of-thought (CoT) reasoning (Wang et al., 2025b), inspired by the success of OpenAI-o1 (Jaech et al., 2024). Several recent works have demonstrated this capability,

---

[*]First Authors; [†]Corresponding Authors.

including Mellow (Deshmukh et al., 2025), Audio-CoT (Ma et al., 2025a), and Audio-Reasoner (Xie et al., 2025), which integrate step-by-step reasoning into audio understanding tasks. The second focus area involves leveraging reinforcement learning (RL) techniques to enhance the performance of LALMs. For instance, Qwen2-Audio (Chu et al., 2024) incorporates Direct Preference Optimization (DPO) (Rafailov et al., 2023) to align models with human preferences. Group Relative Policy Optimization (GRPO), introduced by DeepSeek (Shao et al., 2024), has also gained attention in recent research. R1-AQA (Li et al., 2025) applies GRPO to Qwen2-Audio using the 40k Audio-Visual Question Answering (AVQA) dataset (Yang et al., 2022), while Omni-R1 (Rouditchenko et al., 2025) scales RL to 1,700 steps by constructing 170k multiple-choice questions based on VG-GSound (Chen et al., 2020) and fine-tuning Qwen2.5-Omni (Xu et al., 2025) using GRPO. Some recent works have combined both research directions. SARI (Wen et al., 2025) constructs 32k multiple-choice questions with chain-of-thought annotations and employs Curriculum-Guided RL after supervised fine-tuning (SFT). Step-Audio2 (Wu et al., 2025a) implements multi-stage RL following cold-start initialization: first applying Proximal Policy Optimization (PPO) (Schulman et al., 2017) to enhance reasoning efficiency for real-time audio engagement, then utilizing GRPO to improve the model's audio perceptual capabilities. Audio-Thinker (Wu et al., 2025b) enables hybrid reasoning capabilities through custom-designed rewards and large language model supervision.

However, several challenges persist in current studies. Despite utilizing more data, two-stage paradigms (e.g., SFT followed by RL) may not consistently outperform single-stage post-training, which imposes an effective upper bound on the data volume for post-training. Research on two-stage post-training for LALMs remains limited, which may be partly attributed to the scarcity of large-scale, high-quality datasets specifically designed for LALM post-training. To address these challenges, two primary contributions are made in our work. First, we present AudioMCQ, a high-quality audio multiple-choice question (MCQ) dataset comprising 571k samples, with each sample containing both structured and unstructured CoT annotations, to facilitate improved research in LALM post-training. Second, during the dataset construction, we investigate the tendency for LALMs to derive correct answers solely from textual information without processing the corresponding audio content. We categorize cases of audio-contribution into two types: Explicit Logical Reasoning and Implicit Knowledge Retrieval. Based on this observation, we perform Audio-Contribution Filtering (ACF) by categorizing our data based on the criterion "whether the question can be answered correctly without listening to the audio," resulting in weak audio-contribution and strong audio-contribution splits. Through this filtering approach, we discover simple yet effective post-training paradigms: **Weak-to-Strong**, which involves SFT on weak audio-contribution data followed by GRPO on strong audio-contribution data, and **Mixed-to-Strong**, which involves SFT on mixed audio-contribution data followed by GRPO on strong audio-contribution data. Using AudioMCQ, we achieve first place globally in the DCASE 2025 Audio-Question-Answering challenge (Yang et al., 2025b). Additionally, leveraging our dataset with different training strategies, we fine-tune Qwen2.5-Omni and establish new state-of-the-art (SOTA) performance across multiple benchmarks. Specifically, the Weak-to-Strong strategy achieves 78.2% on MMAU-test-mini and 75.6% on MMAU (Sakshi et al., 2025) [1], while the Mixed-to-Strong strategy attains 67.0% on MMAR (Ma et al., 2025b) and 71.7% on MMSU (Wang et al., 2025a). Both approaches yield significant performance improvements, and we find that the effectiveness of each method on different benchmarks correlates with the inherent audio-contribution characteristics of downstream tasks.

This paper is organized as follows: Section 2 presents the AudioMCQ dataset construction pipeline. Section 3 investigates the zero audio-contribution phenomenon in LALMs and introduces Audio-Contribution Filtering to partition data into weak and strong audio-contribution subsets. Section 4 demonstrates experimental validation of our dataset quality and proposes two effective post-training paradigms: Weak-to-Strong and Mixed-to-Strong training approaches. Finally, we conclude with a discussion of our findings and their implications for future LALM research in Section 5.

## 2 AUDIOMCQ DATASET

The overview of dataset construction is illustrated in Figure 1. Our data source selection follows two principles: (1) avoiding LALMs in source dataset construction to prevent hallucinations introduced by their reliance on textual queries, and (2) ensuring accuracy through a human-verified or model-

---

[1]All MMAU-test-mini and MMAU evaluations in this paper use version 05.15.25.

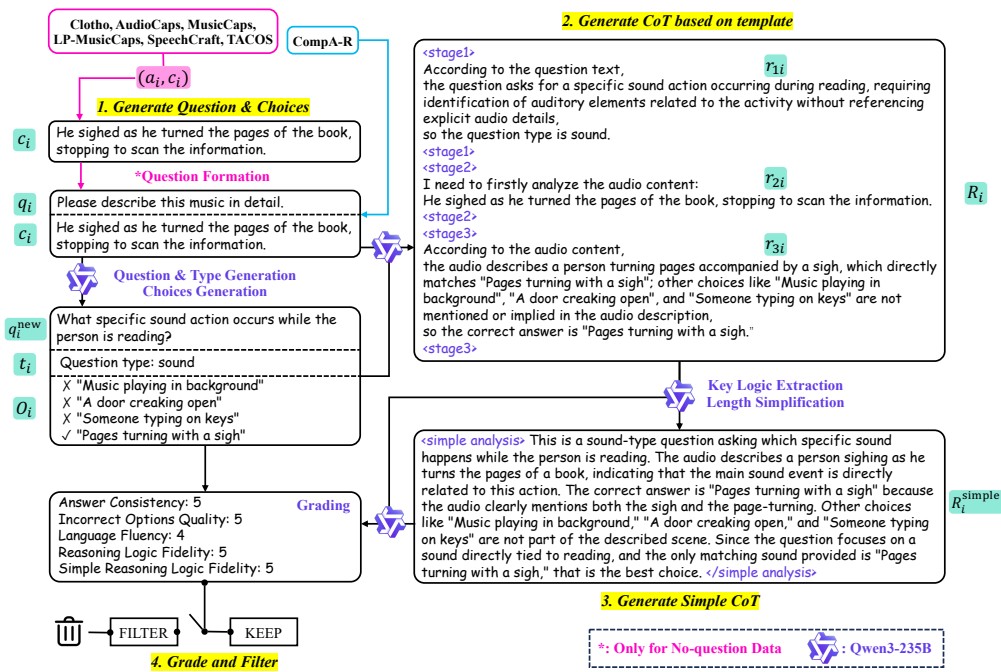

Figure 1: Overview of dataset construction. Detailed prompts are provided in Appendix B. Information on the in-pipeline quality check is provided in Appendix C.

refined pipeline. The training splits from Clotho (Drossos et al., 2020), AudioCaps (v2.0) (Kim et al., 2019), CompA-R (Ghosh et al., 2024), MusicCaps (Agostinelli et al., 2023), LP-MusicCaps (MTT split) (Doh et al., 2023), SpeechCraft (LibriTTS-R split) (Jin et al., 2024), and TACOS (Primus et al., 2025) are used, as summarized in Table 1.

## 2.1 Basic Question-answer Pairs Formation

Six of the seven source datasets contain only audio-caption pairs and lack native question-answer (Q-A) pairs. Therefore, they are transformed into a unified Q-A format for consistent training. Formally, given a source dataset $\mathcal{D}_{\mathrm{src}} = \{(a_i, c_i)\}_{i=1}^{K}$ where $a_i$ denotes audio samples, $c_i$ represents corresponding captions or annotations, and $K$ is the number of samples, the transformation process converts all datasets into a unified format of $\mathcal{D}_{\mathrm{unified}} = \{(a_i, q_i, c_i)\}_{i=1}^{K}$ where $q_i$ denotes the generated question for sample $i$.

The transformation process applies different functions for different datasets. For captioning datasets, the transformation function $f_{\mathrm{caption}} : (a_i, c_i) \rightarrow (a_i, q_{\mathrm{template}}, c_i)$ is applied, where $q_{\mathrm{template}}$ represents one of five predefined question templates described as follows: "Please describe this audio in detail"

Table 1: Overview of our source datasets. AC: Audio Captioning, AQA: Audio Question Answering, MC: Music Captioning, SD: Speech Description, SED: Sound Event Detection.

| Dataset | Samples | Type | Annotation | Preprocessing for Each Sample |
|---|---|---|---|---|
| Clotho | 3,839 | AC | Human | Select longest caption |
| AudioCaps | 91,254 | AC | Human | No preprocessing |
| CompA-R | 198,648 | AQA | Pipeline | No preprocessing |
| MusicCaps | 2,649 | MC | Human | No preprocessing |
| LP-MusicCaps | 15,626 | MC | Pipeline | Use summary-style caption |
| SpeechCraft | 228,944 | SD | Pipeline | Select 3-30s clips |
| TACOS | 10,358 | SED | Human | Convert tags with timestamps to caption |

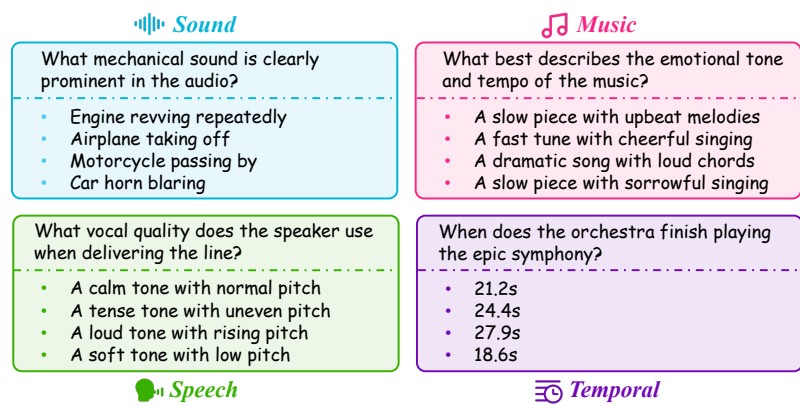

Figure 2: Randomly sampled questions from four distinct question types.

for AudioCaps and Clotho, "Please describe this music in detail" for MusicCaps and LP-MusicCaps, "Please describe this speech in detail" for SpeechCraft, and "Please identify and describe all sound events" for TACOS. For CompA-R, the identity transformation $f_{\text{CompA-R}} : (a_i, q_i, c_i) \rightarrow (a_i, q_i, c_i)$ is applied, as the original question-answer pairs are preserved without modification.

## 2.2 MULTIPLE CHOICE QUESTION CONSTRUCTION

To construct diverse question-answer pairs with varied formulations, a transformation $g : \mathcal{D}_{\text{unified}} \rightarrow \mathcal{D}_{\text{MCQ}}$ is applied using Qwen3-235B (Yang et al., 2025a). For each tuple $(a_i, q_i, c_i) \in \mathcal{D}_{\text{unified}}$, the function generates:

$$g(a_i, q_i, c_i) = (a_i, q_i^{\text{new}}, c_i, O_i, y_i, t_i),\qquad(1)$$

where $q_i^{\text{new}}$ represents the newly constructed question, $c_i$ is the preserved audio content description, $O_i = \{o_1, o_2, o_3, o_4\}$ denotes the set of four answer options with three distractors and one correct answer, $y_i \in \{1, 2, 3, 4\}$ indicates the index of the correct answer, and $t_i \in \mathcal{T}$ represents the question type classification.

For datasets $\mathcal{D}_{\text{flexible}} = \{\text{Clotho, AudioCaps, CompA-R, MusicCaps, LP-MusicCaps, SpeechCraft}\}$, which do not contain event timestamps but provide longer captions, the question type space is defined as $\mathcal{T}_{\text{flexible}} = \{\text{Sound, Music, Speech}\}$ to preserve data diversity. The model automatically determines the appropriate question type based on $q_i^{\text{new}}$ and $O_i$, with one question-answer pair generated per audio sample. For TACOS, the question type space is constrained to $\mathcal{T}_{\text{TACOS}} = \{\text{Temporal}\}$, focusing on temporal-related questions (e.g., event sequences, timestamps), and the number of generated question-answer pairs per audio sample varies from 1 to 4, as determined by the model.

## 2.3 CHAIN-OF-THOUGHT GENERATION

Previous studies show that CoT improves language model performance. To support CoT research for LALMs, we introduce a structured CoT generation component. A three-stage CoT generation function $h : \mathcal{D}_{\text{MCQ}} \rightarrow \mathcal{D}_{\text{candidate}}$ is defined, where $\mathcal{D}_{\text{candidate}}$ represents the dataset at the final stage before quality inspection. For each multiple choice question tuple $(a_i, q_i^{\text{new}}, c_i, O_i, y_i, t_i)$, the function generates:

$$h(a_i, q_i^{\text{new}}, c_i, O_i, y_i, t_i) = (a_i, q_i^{\text{new}}, O_i, y_i, t_i, R_i, R_i^{\text{simple}}),\qquad(2)$$

where $R_i = (r_{1i}, r_{2i}, r_{3i})$ represents the structured three-stage reasoning process:

$$r_{1i} = \text{QuestionTypeAnalysis}(q_i^{\text{new}}, t_i),\qquad(3)$$
$$r_{2i} = \text{AudioContentAnalysis}(c_i),\qquad(4)$$
$$r_{3i} = \text{AnswerSelection}(r_{1i}, r_{2i}, O_i, y_i).\qquad(5)$$

To benefit research on more efficient reasoning, a simplification function $s : R \rightarrow R^{\text{simple}}$ is then applied using Qwen3-235B, converting the three-stage structured CoT $R$ into natural language unstructured CoT $R^{\text{simple}}$, where $|R^{\text{simple}}| < |R|$ in terms of reasoning chain length.

The generated candidate dataset is thus defined as:

$$\mathcal{D}_{\text{candidate}} = \{(a_i, q_i^{\text{new}}, O_i, y_i, t_i, R_i, R_i^{\text{simple}})\}_{i=1}^{M}, \tag{6}$$

where $M$ represents the total number of generated multiple choice questions.

## 2.4 QUALITY CONTROL AND FILTERING

To ensure high-quality audio-based questions, each sample $\{(a_i, q_i^{\text{new}}, O_i, y_i, t_i, R_i, R_i^{\text{simple}})\}_{i=1}^{M} \in \mathcal{D}_{\text{candidate}}$ is sent to Qwen3-235B for automatic evaluation across five quality dimensions: answer consistency, distractor quality, language fluency, reasoning logic, and simplified reasoning quality. Each dimension is scored on a 5-point scale. Samples where any of the five scores falls below 4 are filtered out, resulting in the final dataset $\mathcal{D}_{\text{final}}$, constituting the AudioMCQ dataset with 571,118 data samples. The question type distribution and source data distribution of AudioMCQ are presented in Figure 3 (a) and Figure 3 (b), respectively. Our analysis reveals that speech-related questions constitute the largest proportion at 47.0% (268,299 samples), followed by sound-based questions at 39.1% (223,126 samples). Music-related questions account for 8.1% (46,373 samples), while temporal questions represent 5.8% (33,320 samples). Regarding the source data distribution, SpeechCraft is the largest contributor, providing 39.9% (228,033 samples) of the total dataset. CompA-R follows as the second-largest source, contributing 34.5% (197,218 samples). AudioCaps accounts for 15.8% (90,549 samples), while TACOS, LP-MusicCaps, Clotho, and MusicCaps contribute 5.8% (33,320 samples), 2.7% (15,560 samples), 0.7% (3,801 samples), and 0.5% (2,637 samples), respectively.

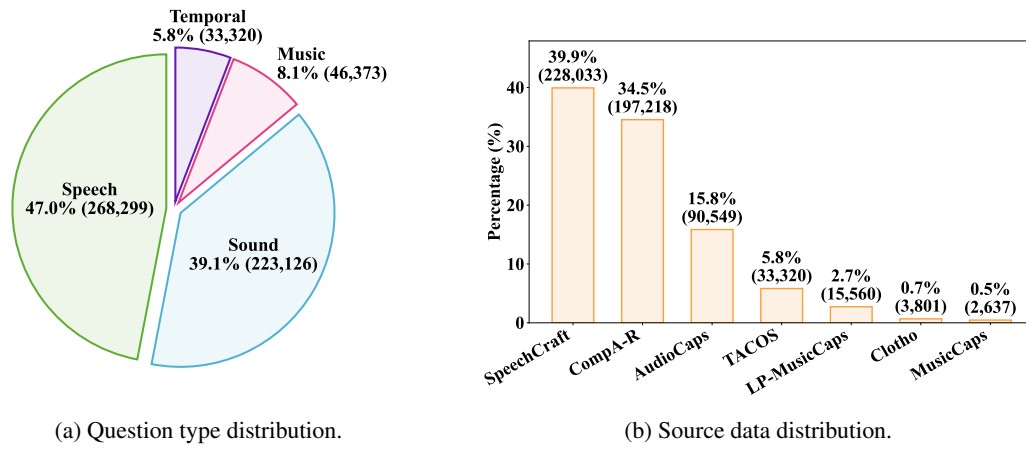

(a) Question type distribution.            (b) Source data distribution.

Figure 3: Distribution analysis of AudioMCQ dataset.

# 3 WHEN LALMS SKIP AUDIO: EXPLORING AUDIO-CONTRIBUTION

## 3.1 DEFINITION OF AUDIO-CONTRIBUTION

We define **audio-contribution** as the degree to which audio features contribute to a LALM's ability to correctly answer a question within a given audio-question pair. Formally, for a multiple-choice question with audio $a_i$, question $q_i$, options $O_i$, and ground truth answer $y_i$, let $\hat{y}(a_i, q_i, O_i)$ denote the model prediction when the audio, question, and options are provided, and $\hat{y}(\mathbf{0}, q_i, O_i)$ represent the prediction using only textual information with a 30-second silent audio $\mathbf{0}$ replacing the original audio. Unlike MMAU or RUListening (Zang et al., 2025) which uses Gaussian noise as replacement, we use silent audio to isolate textual reasoning. The audio-contribution $\mathcal{AC}(a_i, q_i, O_i)$ is defined as:

$$\mathcal{AC}(a_i, q_i, O_i) = \mathbb{I}[\hat{y}(a_i, q_i, O_i) = y_i] - \mathbb{I}[\hat{y}(\mathbf{0}, q_i, O_i) = y_i], \tag{7}$$

where $\mathbb{I}[\cdot]$ is the indicator function that returns 1 if the condition is true and 0 otherwise. Conversely, we introduce the concept of **zero audio-contribution**, which occurs when $\mathcal{AC}(a_i, q_i, O_i) = 0$. This indicates that LALMs produce identical predictions regardless of audio availability:

$$\hat{y}(a_i, q_i, O_i) = \hat{y}(\mathbf{0}, q_i, O_i). \tag{8}$$

Table 2: Performance breakdown of LALMs across audio benchmarks **with silent audio input**.

| **MMAU-test-mini Performance by Subset (%)** | | | | | | | |
|---|---|---|---|---|---|---|---|
| **Subset** | **Qwen2-Audio** | **A-Flamingo2** | **R1-AQA** | **Kimi-Audio** | **Qwen2.5-Omni** | **Average** | **Random Guess** |
| Sound | 42.0 | 56.5 | 56.2 | 67.6 | 51.7 | **54.8** | 25.0 |
| Music | 39.8 | 62.3 | 49.7 | 57.2 | 50.9 | **52.0** | 25.0 |
| Speech | 34.5 | 41.4 | 44.1 | 50.5 | 42.9 | **42.7** | 26.7 |
| **Overall** | **38.8** | **53.4** | **50.0** | **58.4** | **48.5** | **49.8** | **25.5** |
| **MMAR Performance by Subset (%)** | | | | | | | |
| **Subset** | **Qwen2-Audio** | **A-Flamingo2** | **R1-AQA** | **Kimi-Audio** | **Qwen2.5-Omni** | **Average** | **Random Guess** |
| Perception | 30.9 | 30.5 | 36.1 | 39.1 | 27.7 | **32.9** | 27.2 |
| Semantic | 34.5 | 39.6 | 37.6 | 40.5 | 35.4 | **37.5** | 31.4 |
| Signal | 41.9 | 44.2 | 37.2 | 53.5 | 48.8 | **45.1** | 33.0 |
| Cultural | 32.6 | 31.9 | 30.5 | 40.4 | 31.9 | **33.5** | 28.4 |
| **Overall** | **33.1** | **35.0** | **36.0** | **46.5** | **32.4** | **36.6** | **29.3** |
| **MMSU Performance by Subset (%)** | | | | | | | |
| **Subset** | **Qwen2-Audio** | **A-Flamingo2** | **R1-AQA** | **Kimi-Audio** | **Qwen2.5-Omni** | **Average** | **Random Guess** |
| Perception | 30.1 | 28.3 | 42.0 | 29.4 | 26.0 | **31.2** | 25.0 |
| Reasoning | 40.2 | 43.7 | 43.3 | 53.4 | 42.8 | **44.7** | 25.0 |
| **Overall** | **35.3** | **35.8** | **42.7** | **41.0** | **34.1** | **37.8** | **25.0** |

To investigate this phenomenon systematically, we analyze several mainstream open-source LALMs across three established benchmarks: MMAU-test-mini, MMAR, and MMSU. Specifically, we evaluate existing LALMs on these benchmarks by replacing the audio component in each audio-question pair with silent audio input $\mathbf{0}$. The detailed breakdown of this analysis is presented in Table 2. On MMAU-test-mini, models achieve an overall average of 49.8% accuracy with silent audio input, compared to the random guess baseline of 25.5%. Sound-based queries show the highest accuracy with silent audio input at 54.8%, followed by music (52.0%) and speech (42.7%). MMAR demonstrates an overall accuracy of 36.6% with silent audio input, with signal processing achieving the highest accuracy at 45.1%. On MMSU, the overall accuracy with silent audio input reaches 37.8%, with reasoning tasks showing higher performance (44.7%) than perception tasks (31.2%). Overall, all benchmarks exhibit pronounced zero audio-contribution phenomena, with MMAR and MMSU showing relatively lower accuracies with silent audio input, while MMAU-test-mini demonstrates higher accuracy with silent audio input, reaching approximately 50%.

## 3.2 AUDIO-CONTRIBUTION FILTERING

Based on these preliminary insights and experimental findings on audio-contribution, we design Audio-Contribution Filtering (ACF) to partition the AudioMCQ dataset into two distinct subsets: weak audio-contribution and strong audio-contribution. Specifically, three LALMs—Audio-Flamingo2 (A-Flamingo2) (Ghosh et al., 2025), R1-AQA, and Kimi-Audio (Ding et al., 2025)—are employed to evaluate questions within AudioMCQ, with the audio component in each audio-question pair being replaced with 30 seconds of silent audio.

Let $\mathcal{M} = \{M_1, M_2, M_3\}$ denote the set of three evaluation models, where $M_1$ is A-Flamingo2, $M_2$ is R1-AQA, and $M_3$ is Kimi-Audio. For a given sample $(a_i, q_i, O_i, y_i) \in \mathcal{D}_{\text{final}}$, let $y_j(\mathbf{0}, q_i, O_i)$ denote the prediction of the model $M_j$ when provided with silent audio. We define the correctness indicator for each model as:

$$\mathcal{C}_j(q_i, O_i, y_i) = \mathbb{I}[y_j(\mathbf{0}, q_i, O_i) = y_i]. \tag{9}$$

If at least two out of three models can correctly answer the question under silent audio input $\mathbf{0}$, the data sample is assigned to weak audio-contribution subset $\mathcal{D}_{\text{weak}}$; otherwise, it is classified into strong audio-contribution subset $\mathcal{D}_{\text{strong}}$:

$$\mathcal{ACF}(q_i, O_i, y_i) = \begin{cases} \text{Weak} & \text{if } \sum_{j=1}^{3} \mathcal{C}_j(q_i, O_i, y_i) \geq 2, \\ \text{Strong} & \text{otherwise}, \end{cases} \tag{10}$$

$$\mathcal{D}_{\text{weak}} = \{(a_i, q_i, O_i, y_i) : \mathcal{ACF}(q_i, O_i, y_i) = \text{Weak}\}, \tag{11}$$

$$\mathcal{D}_{\text{strong}} = \{(a_i, q_i, O_i, y_i) : \mathcal{ACF}(q_i, O_i, y_i) = \text{Strong}\}. \tag{12}$$

Table 3: Performance of LALMs on different audio datasets **with silent audio input** and the audio-contribution split ratios of these datasets. AC refers to audio-contribution.

| Source Dataset | Samples | Model Accuracy w/o Audio (%) | | | AC Split Ratio(%) | |
|---|---|---|---|---|---|---|
| | | A-Flamingo2 | R1-AQA | Kimi-Audio | Weak | Strong |
| Clotho | 3,801 | 44.9 | 40.7 | 58.3 | 47.4 | 52.6 |
| AudioCaps | 90,549 | 41.7 | 38.2 | 59.0 | 44.9 | 55.1 |
| CompA-R | 197,218 | 69.8 | 64.7 | 81.6 | 75.5 | 24.5 |
| MusicCaps | 2,637 | 48.7 | 41.5 | 54.6 | 46.8 | 53.2 |
| LP-MusicCaps | 15,560 | 47.5 | 41.9 | 60.6 | 49.2 | 50.8 |
| SpeechCraft | 228,033 | 35.0 | 47.1 | 58.4 | 45.6 | 54.4 |
| TACOS | 33,320 | 31.2 | 34.1 | 35.4 | 26.7 | 73.3 |
| **Overall** | **571,118** | **48.3** | **50.8** | **65.2** | **54.8** | **45.2** |

Table 4: Performance of LALMs on different audio understanding benchmarks **with silent audio input** and the audio-contribution split ratios of these benchmarks. AC refers to audio-contribution.

| Source Dataset | Samples | Model Accuracy (%) | | | AC Split Ratio(%) | |
|---|---|---|---|---|---|---|
| | | A-Flamingo2 | R1-AQA | Kimi-Audio | Weak | Strong |
| MMAU-test-mini | 1000 | 53.4 | 50.0 | 58.4 | 53.9 | 46.1 |
| MMAR | 1000 | 35.0 | 36.0 | 40.5 | 32.9 | 67.1 |
| MMSU | 5000 | 35.8 | 42.7 | 41.0 | 35.7 | 64.3 |

Table 3 reveals significant variations in audio-contribution distribution across different datasets, reflecting their inherent characteristics and design objectives. TACOS demonstrates the highest proportion of strong audio-contribution samples (73.3%), which aligns with its nature as a temporal reasoning dataset requiring precise temporal understanding and sequential audio analysis. In contrast, CompA-R exhibits the highest proportion of weak audio-contribution samples (75.5%), reflecting its focus on compositional reasoning tasks. We also apply the same ACF methodology to three additional audio understanding benchmarks: MMAU-test-mini, MMAR, and MMSU. Subsequently, we refer to their strong audio-contribution subsets as MMAU-test-mini-ACstrong, MMAR-ACstrong, and MMSU-ACstrong, respectively. These filtered benchmarks provide a more rigorous evaluation framework for assessing models' genuine audio comprehension capabilities. Table 4 reveals significant variations in audio-contribution distribution across different benchmarks, reflecting their inherent characteristics and design objectives. MMAU-test-mini exhibits the highest proportion of weak audio-contribution samples (53.9%), while MMAR demonstrates the highest proportion of strong audio-contribution samples (67.1%).

## 3.3 CASES OF ZERO AUDIO-CONTRIBUTION

Zero audio-contribution cases are notably prevalent in both our constructed AudioMCQ dataset and benchmark evaluations. Therefore, we investigate the underlying causes and identify two distinct patterns. The first type, **Explicit Logical Reasoning**, occurs when the model can directly infer the correct answer from textual cues present in the question. The second type is **Implicit Knowledge Retrieval**, where the model relies on knowledge acquired during training to identify the correct option despite lacking explicit textual hints. Examples of both types can be found in Appendix D. To investigate the distribution of these types, each item is analyzed using Qwen3-235B by providing the question text, options, and the correct answer to determine whether the answer can be inferred without audio input. Among the 313,177 samples in the weak audio-contribution split, 31.1% (97,364 samples) are classified as Explicit Logical Reasoning, of which 85.1% (82,814 samples) are derived from the CompA-R dataset designed for audio-based reasoning, while Implicit Knowledge Retrieval comprises 68.9% (215,813 samples).

Table 5: Performance comparison across different models.[2]

| Method | MMAU-test-mini | MMAU | MMAR | MMSU |
|---|---|---|---|---|
| Audio-Reasoner | 67.7 | 63.8 | 36.8 | 49.2 |
| R1-AQA | 68.9 | 68.5 | 50.8 | 61.6 |
| Kimi-Audio | 68.2 | 64.4 | 57.6 | 59.3 |
| SARI | 67.0 | – | – | 66.0 |
| Qwen2.5-Omni (backbone) | 71.5 | 71.0 | 56.7 | 60.6 |
| Audio Flamingo 3 | 73.3 | 72.4 | 60.1 | 62.3 |
| Omni-R1 | 77.0 | 75.0 | 63.4 | – |
| Audio-Thinker | 78.0 | 75.4 | 65.3 | – |
| GPT4o-Audio | 62.5 | 60.8 | 63.5 | 56.4 |
| Gemini-2.0-Flash | 70.5 | 67.0 | 65.6 | 51.0 |
| *Our Methods* | | | | |
| All Data SFT | 75.2 | 75.0 | 64.6 | 64.0 |
| All Data GRPO | 78.1 | 75.4 | 63.0 | 70.2 |
| Mix AC SFT + Mix AC GRPO | 74.2 | 74.4 | 64.9 | 69.2 |
| Weak AC SFT + Strong AC GRPO | **78.2** | **75.6** | 65.3 | 69.3 |
| Mix AC SFT + Strong AC GRPO | 76.4 | 75.1 | **67.0** | **71.7** |

## 4 EXPERIMENTS

We conduct a series of training experiments using Qwen2.5-Omni as the backbone model to validate our dataset quality and training paradigms. Given our focus on perceptual fidelity rather than reasoning, CoT is excluded from the following evaluations.

### 4.1 DATASET QUALITY VALIDATION

To validate the quality of the constructed AudioMCQ dataset, we perform two experiments: SFT training and GRPO training using all data from AudioMCQ.

GRPO is a novel RL approach that eliminates the need for a separate value function approximation as required in PPO. Instead of using a value function as the baseline, GRPO leverages the average reward of multiple sampled outputs produced in response to the same question as the baseline. Specifically, for each question $q$, GRPO samples a group of outputs $\{o_1, o_2, \cdots, o_G\}$ from the old policy $\pi_{\theta_{old}}$ and optimizes the policy model by maximizing the following objective:

$$\mathcal{J}_{GRPO}(\theta) = \mathbb{E}[q \sim P(Q), \{o_i\}_{i=1}^{G} \sim \pi_{\theta_{old}}(O|q)]$$

$$\frac{1}{G} \sum_{i=1}^{G} \frac{1}{|o_i|} \sum_{t=1}^{|o_i|} \left\{ \min \left[ \frac{\pi_\theta(o_{i,t}|q, o_{i,<t})}{\pi_{\theta_{old}}(o_{i,t}|q, o_{i,<t})} \hat{A}_{i,t}, \text{clip} \left( \frac{\pi_\theta(o_{i,t}|q, o_{i,<t})}{\pi_{\theta_{old}}(o_{i,t}|q, o_{i,<t})}, 1 - \epsilon, 1 + \epsilon \right) \hat{A}_{i,t} \right] - \beta D_{KL}[\pi_\theta || \pi_{ref}] \right\}, \quad (13)$$

where $\theta$ represents the parameters of the policy model being optimized, $G$ is the group size (number of sampled outputs per question), $|o_i|$ denotes the length of the $i$-th output, $o_{i,t}$ is the $t$-th token in the $i$-th output, $o_{i,<t}$ represents all tokens before position $t$ in the $i$-th output, $\epsilon$ is the clipping parameter, $\beta$ is the KL divergence (van Erven & Harremos, 2014) regularization coefficient, $\pi_{ref}$ is the reference policy, and $\hat{A}_{i,t}$ is the advantage calculated based on relative rewards of the outputs inside each group only.

In the experiments, the optimal checkpoint is determined based on performance on MMAU-test-mini-4k as validation. This expanded version of MMAU-test-mini is created by replicating each question to ensure that the correct option appears in each position at least once. The selected checkpoint is subsequently evaluated on MMAU-test-mini, MMAU, MMAR, and MMSU. After conducting 2000 steps of SFT, we achieve 75.2% on MMAU-test-mini, 75.0% on MMAU and 64.6% on MMAR. After conducting 1200 steps of GRPO, 78.1% and 75.4% are achieved on MMAU-test-mini and MMAU respectively, outperforming all previous models. Additionally, 70.2% is achieved on MMSU, marking the first time this benchmark score exceeds 70%, representing a 6.2% improvement over the previous SFT result of 64%. These results demonstrate the effectiveness and high quality of our AudioMCQ dataset.

---

[2]Performance scores are sourced from official benchmarks and websites when available; otherwise, we reproduce results for open-source models or use reported scores from official papers for unreleased models.

## 4.2 SFT-TO-RL TRAINING PARADIGMS

Although the previous SFT-only and RL-only experiments in Section 4.1 demonstrate promising results, there is still room for improvement. A critical open question is how to optimally partition data between SFT and RL stages. To address this challenge, we explore the possibility of allocating SFT and RL data based on audio-contribution levels. We design additional SFT-to-RL experiments to investigate this approach. First, a baseline is established, namely **Mixed-to-Mixed**, which applies SFT on mixed audio-contribution data followed by GRPO on mixed audio-contribution data, with both stages randomly sampled from AudioMCQ. Subsequently, two distinct training paradigms are designed: **Weak-to-Strong**, which employs SFT on weak audio-contribution data followed by GRPO on strong audio-contribution data, and **Mixed-to-Strong**, which uses mixed audio-contribution data SFT followed by strong audio-contribution data GRPO. To ensure fair comparison, all experiments maintain consistent data volumes with 313,177 samples for SFT, non-overlapping SFT/GRPO data. The optimal checkpoint selection based on MMAU-test-mini-4k performance, shown in Figure 4.

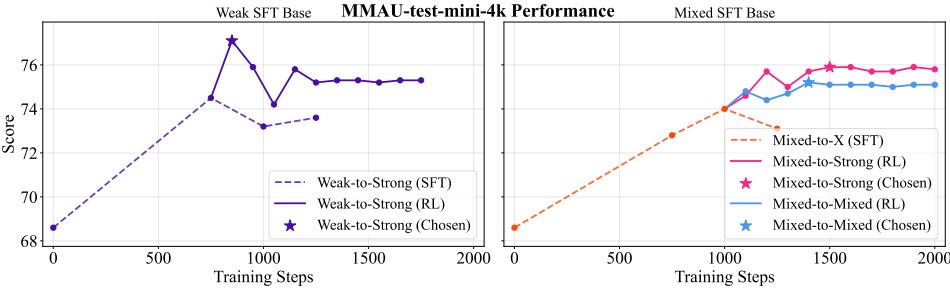

Figure 4: Performance comparison of three training approaches on MMAU-test-mini-4k for optimal checkpoint selection. Note that "Mixed-to-X (SFT)" indicates the shared SFT phase of Mixed-to-Mixed and Mixed-to-Strong approaches.

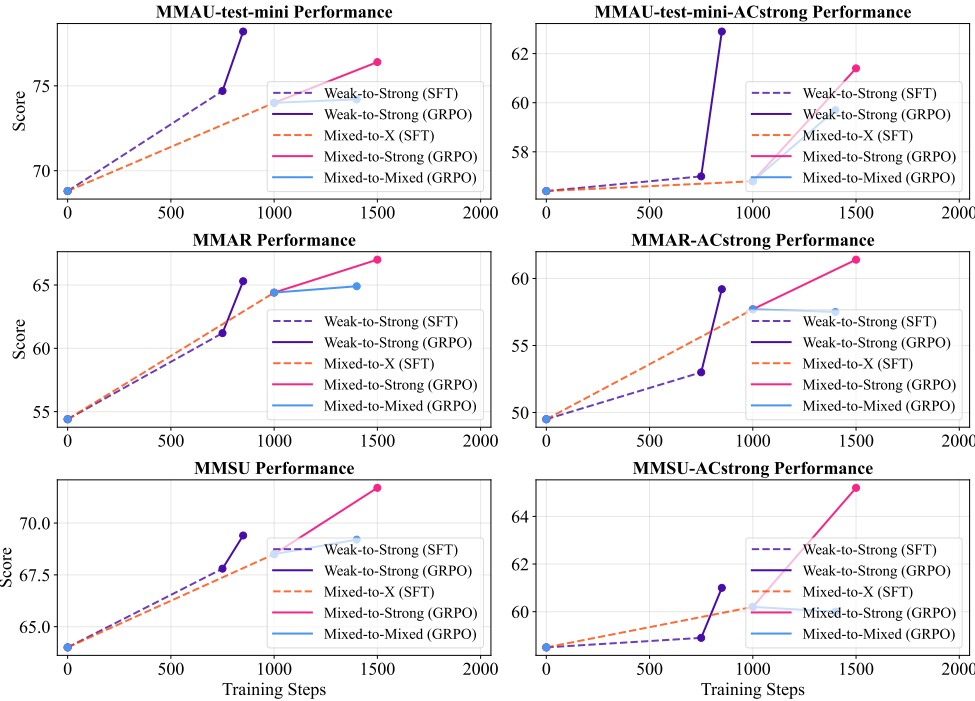

Figure 5: Performance comparison of three training approaches across MMAU-test-mini, MMAR, MMSU and their strong audio-contribution splits. Only the optimal checkpoints are displayed.

The baseline Mixed-to-Mixed approach (1000 steps SFT + 400 steps GRPO with mixed audio-contribution data) achieves 64.9% and 69.2% on MMAR and MMSU respectively, demonstrating suboptimal MMAU performance. In contrast, the Weak-to-Strong paradigm (750 steps weak audio-contribution SFT + 100 steps strong audio-contribution GRPO) surpasses the previous 1200-step GRPO-only approach, achieving SOTA performance with 78.2% on MMAU-test-mini and 75.6% on MMAU, while obtaining 65.3% on MMAR. The Mixed-to-Strong approach (1000 steps mixed audio-contribution SFT + 500 steps strong audio-contribution GRPO) attains SOTA performance on MMAR and MMSU (67.0% and 71.7% respectively).

Although both approaches—**Weak-to-Strong** and **Mixed-to-Strong**—significantly outperform the **Mixed-to-Mixed** baseline, understanding where these methods enhance model capabilities and when to select between them remains important. Therefore, the performance of all three methods is evaluated at three training stages (initial, post-SFT, and post-GRPO) across six benchmarks (MMAU-test-mini, MMAR, MMSU, and their respective strong audio-contribution splits), with results shown in Figure 5. Two key conclusions are derived from this analysis. First, using strong audio-contribution data for RL is important, as GRPO with mixed audio-contribution data yields minimal performance improvement and can even results in degradation of audio-based question-answering capability on the AC-strong splits of MMAR and MMSU. In contrast, GRPO with strong audio-contribution data yields significant improvements across all benchmarks and their AC-strong splits, reflecting a substantial gain in the model's perception capability, particularly pronounced on the strong audio-contribution benchmarks (MMAR and MMSU). Second, SFT data selection should align with specific downstream task characteristics. Since approximately half of MMAU-test-mini questions can be answered without audio input, it represents a relatively weak audio-contribution benchmark, making weak audio-contribution data during SFT training better matched to the benchmark distribution. For MMAR and MMSU, which are relatively strong audio-contribution benchmarks, naturally distributed mixed audio-contribution data for SFT training proves more suitable.

## 5 CONCLUSION

We first present AudioMCQ, a comprehensive high-quality audio multiple-choice question dataset comprising 571k samples with structured and unstructured CoT annotations for LALMs post-training. Through systematic investigation, the zero audio-contribution phenomenon is identified, where current LALMs frequently derive correct answers from textual information alone with silent audio input, with rates reaching 49.8% on MMAU, 36.6% on MMAR, and 37.8% on MMSU benchmarks. Audio-Contribution Filtering methodology is developed to partition the data into weak (54.8%) and strong (45.2%) audio-contribution subsets based on whether a question is correctly answered by more than half of the models under silent audio input conditions. Within weak audio-contribution data, two distinct patterns are identified: Explicit Logical Reasoning (31.1%) and Implicit Knowledge Retrieval (68.9%). Based on these insights, two effective post-training paradigms are developed that achieve SOTA performance among open-source models. The Weak-to-Strong approach attains 78.2% on MMAU-test-mini and 75.6% on MMAU, while Mixed-to-Strong reaches 67.0% on MMAR and 71.7% on MMSU. The methodology demonstrates that RL with strong audio-contribution data is important for capability enhancement, while SFT data selection should be aligned with downstream task characteristics. These contributions establish new benchmarks for LALM post-training and provide practical guidance for data allocation strategies. Future work could focus on establishing a more robust audio-contribution-based data partitioning for AudioMCQ, extending Audio-Contribution Filtering to general audio question-answering tasks, and further investigating CoT reasoning capabilities in audio understanding contexts.

## 6 ACKNOWLEDGEMENTS

This work was supported by the NSFC Young Scientists Fund (Category C) [grant number 62501512], the Ant Group Research Intern Program, and the Engineering and Physical Sciences Research Council (EPSRC) [grant numbers EP/T019751/1, EP/Y028805/1].

## 7 ETHICS STATEMENT

We have no ethical concerns to declare regarding this work.

## 8 REPRODUCIBILITY STATEMENT

We have made efforts to ensure the reproducibility of our work. Implementation details, hyperparameters, and evaluation protocols are provided in the main paper and Appendix.

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

## A   USE OF LARGE LANGUAGE MODELS

Large Language Models are used solely as part of the AudioMCQ creation pipeline and for polishing portions of the manuscript. All research concepts, methodology, experimental design, analysis, and conclusions represent original work by the authors.

## B   PROMPTS

### B.1   MULTIPLE-CHOICE QUESTION GENERATION (SOUND, MUSIC AND SPEECH)

```
1   You are a test designer for advanced audio comprehension exams. Your task is to generate a new multiple-choice
     question BASED ON an existing question-answer pair, designed to assess understanding that can ONLY be gained
     by listening to the audio.
2
3   Here is the source information:
4   ORIGINAL QUESTION: {q_text}
5   ORIGINAL ANSWER: {answer}
6
7   Use the following six requirements to create your JSON output:
8
9   REQUIREMENT 1: QUESTION TYPE
10  Choose ONE of: "sound", "music", or "speech" based on what the new question targets.
11
12  REQUIREMENT 2: NEW QUESTION
13  Write a new question that:
14  - Ends with a question mark
15  - Is clear, concise, and **easy to be understood**
16  - Tests specific content that definitely *exists in the ORIGINAL ANSWER*
17  - Focus on *audio-dependent understanding* that cannot be inferred from text alone
18
19  REQUIREMENT 3: MULTIPLE-CHOICE OPTIONS
20  Write 4 options (3 incorrect options and 1 correct answer), each between 1-8 words. Guidelines:
21  - Use capital letters only at the start, no ending punctuation
22  - All four options must maintain consistent levels of detail
23  - All four options must follow the same sentence structure and grammar type
24  - All four options must appear equally plausible to a partially attentive listener
25  - All four options must have CONSISTENT word count to prevent identifying correct answers by length
26
27  REQUIREMENT 4: CORRECT ANSWER DESIGN
28  The correct answer must:
29  - Require *actual listening to the audio* to answer
30  - Be based on content that definitely *exists in the ORIGINAL ANSWER*
31  - Not reuse exact phrases from the original answer
32  - Extract only the most essential information while omitting secondary details
33  - REPHRASE the content using synonyms or parallel expressions
34
35  REQUIREMENT 5: DISTRACTOR DESIGN
36  Each incorrect option should:
37  - Reflect plausible misunderstandings or shallow interpretations
38  - Match the correct answer's structure, complexity, and vocabulary level
39  - Focus on the SAME TOPIC/ASPECT as the correct answer to maintain thematic consistency
40  - Have consistent length and grammatical structure as the correct answer
41  - Be clearly distinct from the correct answer
42
43  REQUIREMENT 6: VALIDATION CHECK
44  Ensure:
45  - No answer is significantly longer or more detailed
46  - No answer stands out due to word choice, phrasing, or tone
47  - The question-answer pair tests comprehension of audio, not logic
48  - The question cannot be solved by a language model without audio input
49
50  IMPORTANT:
51  Return ONLY a JSON object with EXACTLY these keys:
52  {{
53  "new_question_type": "question type",
54  "new_question": "Your new question here?",
55  "correct_answer": "Correct answer option here",
56  "incorrect_options": [
57      "First incorrect option",
58      "Second incorrect option",
59      "Third incorrect option"
60  ]
61  }}
62
63  DO NOT:
64  - Add extra explanation
65  - Include trailing commas
66  - Use quotation marks inconsistently
67  - Make the correct answer noticeably different in length or tone
68
69  CRUCIAL: Your output must be indistinguishable to a reader (or AI) who has not heard the audio. Only someone
     who listened carefully should be able to answer correctly.
70
71  {style_reference}
```

## B.2 MULTIPLE-CHOICE QUESTION GENERATION (TEMPORAL)

```
1   You are a test designer for advanced audio comprehension exams. Your task is to generate a new multiple-choice
     question BASED ON an existing question-answer pair, designed to assess understanding of TEMPORAL INFORMATION
     that can ONLY be gained by listening to the audio.
2
3   Here is the source information:
4   ORIGINAL QUESTION: {q_text}
5   ORIGINAL ANSWER: {answer}
6
7   Use the following six requirements to create your JSON output:
8
9   REQUIREMENT 1: QUESTION TYPE
10  Always use "temporal" as the question type.
11
12  REQUIREMENT 2: NEW QUESTION
13  Write a new question that:
14  - Ends with a question mark
15  - Is clear and concise with no more than 20 words
16  - Is well-formed with no internal punctuation and long word phrases
17  - Tests specific TEMPORAL INFORMATION that definitely *exists in the ORIGINAL ANSWER*
18  - Clearly distinguishes temporal events using sequence indicators or appropriate adjectives/adverbs when the
     audio contains other similar events
19  - Cannot be answered by common knowledge, world knowledge, or logic alone
20
21  REQUIREMENT 3: MULTIPLE-CHOICE OPTIONS
22  Write 4 options (3 incorrect options and 1 correct answer). Guidelines:
23  - No ending punctuation
24  - All four options must follow the same sentence structure
25  - If time values are present, all four options must have DISTINCT time values *at least {random_seconds}s*
     apart from each other
26  - If time values are present, all time values in the options must not exceed 30.0s
27
28  REQUIREMENT 4: CORRECT ANSWER DESIGN
29  The correct answer must:
30  - Require *actual listening to the audio* to capture the correct temporal information
31  - Be based on TEMPORAL INFORMATION that definitely *exists in the ORIGINAL ANSWER*
32
33  REQUIREMENT 5: DISTRACTOR DESIGN
34  Each incorrect option should:
35  - Reflect PLAUSIBLE but incorrect temporal information
36  - Have consistent length and grammatical structure as the correct answer
37  - Show time differences proportional to both the audio's total length and the correct answer's value
38
39  REQUIREMENT 6: VALIDATION CHECK
40  Ensure:
41  - No answer is significantly longer or more detailed
42  - The question-answer pair tests comprehension of temporal information in audio, not logic
43  - The question cannot be solved by a language model without audio input
44
45  IMPORTANT:
46  Return ONLY a JSON object with EXACTLY these keys:
47  {{
48  "new_question_type": "temporal",
49  "new_question": "Your new question here?",
50  "correct_answer": "Correct answer option here",
51  "incorrect_options": [
52      "First incorrect option",
53      "Second incorrect option",
54      "Third incorrect option"
55  ]
56  }}
57
58  DO NOT:
59  - Add extra explanation
60  - Include trailing commas
61  - Use quotation marks inconsistently
62  - Make the correct answer noticeably different in length or tone
63
64  CRUCIAL: Your output must be indistinguishable to a reader (or AI) who has not heard the audio. Only someone
     who listened carefully to the temporal sequence should be able to answer correctly.
65
66  {style_reference}
```

## B.3 CHAIN-OF-THOUGHT GENERATION (STRUCTURED)

```
1   TASK: Complete the following THINKING PROCESS that demonstrates how to arrive at the correct answer for this
     audio-based multiple-choice question.
2
3   QUESTION DETAILS:
4   - Question: "{entry['question_text']}"
5   - Question Type: {entry['question_type']}
6   - Choices: {", ".join([f'"{opt}"' for opt in entry['multi_choice']])}
7   - Correct Answer: "{entry['answer']}"
8
9   INCOMPLETE THINKING PROCESS:
10  According to the question text, <first_analysis>...</first_analysis>, so the question type is {entry[
     'question_type']}.
11
```

```
12  I need to firstly analyze the audio content:
13  {entry['original_answer']}
14
15  According to the audio content, <second_analysis>...</second_analysis>, so the correct answer is "{entry['
    answer']}".
16
17  COMPLETION REQUIREMENTS:
18
19  **FIRST THINKING PROCESS (Question Analysis):**
20
21  1.1 Style Instructions:
22  - Maximum 30 words
23  - Use lowercase for the first word
24  - Write as one CONTINUOUS paragraph without breaks
25  - Be analytical and methodical
26  - Maintain COHERENCE with the surrounding context
27
28  1.2 Content Instructions:
29  1. Identify the key information being asked for
30  2. Clarify what evidence you need to find in the audio
31  3. Connect the question requirements to the question type
32  4. **Do not mention any information in "{entry['original_answer']}".**
33
34  **SECOND THINKING PROCESS (Audio Analysis & Answer Selection):**
35
36  2.1 Style Instructions:
37  - Maximum 30 words
38  - Use lowercase for the first word
39  - Write as one CONTINUOUS paragraph without breaks
40  - Maintain COHERENCE with the surrounding context
41  - Use "quotes" *only when referencing specific answer choices*, not for quoting audio content directly
42
43  2.2 Content Instructions:
44  1. Highlight the key evidence from the audio that directly answers the question
45  2. Explain how this evidence clearly supports the correct answer
46  3. Briefly compare the other available options with the audio content and determine that they are incorrect.
47  4. Make a logical connection from evidence to answer choice
48  5. **Do not mention any sound details or content that are NOT mentioned in the literal content of "{entry['
    original_answer']}".**
49
50  OUTPUT FORMAT:
51  - First thinking process between <first_analysis> and </first_analysis> tags
52  - Second thinking process between <second_analysis> and </second_analysis> tags
```

## B.4  CHAIN-OF-THOUGHT GENERATION (UNSTRUCTURED)

```
1   TASK: Create a SIMPLIFIED thinking process for this audio-based multiple-choice question.
2
3   QUESTION DETAILS:
4   - Question: "{entry['question_text']}"
5   - Question Type: {entry['question_type']}
6   - Choices: {", ".join([f'"{opt}"' for opt in entry['multi_choice']])}
7   - Correct Answer: "{entry['answer']}"
8   - Audio Description: {entry['original_answer']}
9
10  ORIGINAL THINKING PROCESS:
11  {entry.get('thinking_process', '')}
12
13  REQUIREMENTS FOR SIMPLIFIED VERSION:
14
15  1. **Length**: Maximum 150 words total
16  2. **Structure**: Write as one continuous paragraph without breaks
17  3. **Content**:
18     - Briefly identify the question type
19     - Sufficiently introduce and summerize **All of the Sound Event** and **their related information** in
    fluent language
20     - Briefly state the correct answer choice
21     - No need to explain how to eliminate wrong answers, only focus on how to select the correct answer
22
23  4. **Style**:
24     - Use ENGLISH ONLY - absolutely no Chinese characters allowed
25     - Use simple, clear language
26     - Focus on the essential reasoning only
27     - **Self-contained**: Content must be complete and understandable without relying on the ORIGINAL THINKING
    PROCESS
28     - Use **quotation marks** when referencing answer choices
29     - Avoid using expressions like "Event 1", "Event 2", etc.
30
31  OUTPUT FORMAT:
32  Your response must be in the following format:
33  <thinking process>
34  [Your simplified thinking process here]
35  </thinking process>
36
37  Generate the simplified thinking process:
```

## B.5    QUALITY CONTROL AND FILTERING

```
1   TASK: Quality check an audio-based multiple-choice question entry on five specific aspects.
2
3   ENTRY TO EVALUATE:
4   Question: "{question_text}"
5   Multiple Choice Options:
6   {choice_text.strip()}
7   Correct Answer: "{answer}"
8   Original Audio Description: "{original_answer}"
9
10  Thinking Process:
11  {thinking_process}
12
13  Simple Thinking Process:
14  {thinking_process_simple}
15
16  EVALUATION CRITERIA:
17  Please evaluate this entry on the following five aspects, scoring each from 1-5:
18
19  **ASPECT 1: Language Fluency (1-5)**
20  Evaluate the overall language quality, grammar, clarity, and fluency of:
21  - Question text
22  - Multiple choice options
23  - Correct answer
24  - Thinking process
25  - Simple thinking process
26
27  SCORING GUIDE:
28  - **Score 5 (Excellent)**: Perfect grammar, crystal clear expression, natural and fluent language throughout
    all components. Professional quality writing.
29  - **Score 4 (Good)**: Minor grammatical issues or slightly awkward phrasing, but overall clear and
    understandable. Good quality writing.
30  - **Score 3 (Average)**: Some grammatical errors or unclear expressions that may cause confusion, but
    generally comprehensible. Acceptable quality.
31  - **Score 2 (Poor)**: Multiple grammatical errors, unclear or confusing expressions, unnatural language that
    significantly impacts comprehension.
32  - **Score 1 (Very Poor)**: Severe grammatical problems, incomprehensible or highly confusing language,
    extremely poor expression quality.
33
34  **ASPECT 2: Answer Consistency (1-5)**
35  Evaluate how well the correct answer aligns with the information provided in the original audio description.
36
37  SCORING GUIDE:
38  - **Score 5 (Excellent)**: The correct answer perfectly matches and is strongly supported by the original
    audio description. Complete alignment.
39  - **Score 4 (Good)**: The correct answer is well-supported by the audio description with only minor
    discrepancies. Strong alignment.
40  - **Score 3 (Average)**: The correct answer is generally consistent with the audio description but may have
    some unclear connections. Moderate alignment.
41  - **Score 2 (Poor)**: The correct answer has significant inconsistencies with the audio description or lacks
    clear support. Weak alignment.
42  - **Score 1 (Very Poor)**: The correct answer directly contradicts the audio description or is completely
    unsupported by it. No alignment.
43
44  **ASPECT 3: Incorrect Options Quality (1-5)**
45  Evaluate how well the incorrect options are designed - they should NOT appear in or be supported by the
    original audio description.
46  Incorrect Options to evaluate:
47  {incorrect_options_text.strip()}
48
49  SCORING GUIDE:
50  - **Score 5 (Excellent)**: All incorrect options clearly contradict or are completely unsupported by the audio
     description. They are obviously wrong and serve as perfect distractors.
51  - **Score 4 (Good)**: Most incorrect options contradict the audio description, with only minor elements that
    might seem plausible. Good distractors overall.
52  - **Score 3 (Average)**: Some incorrect options contradict the audio description while others may have neutral
     or unclear relationships to it. Mixed quality distractors.
53  - **Score 2 (Poor)**: Many incorrect options are partially supported by or appear in the audio description,
    making them potentially correct. Poor distractors.
54  - **Score 1 (Very Poor)**: Most or all incorrect options are clearly supported by the audio description,
    making them appear correct. Terrible distractors that confuse the question.
55
56  **ASPECT 4: Thinking Process Logic & Fidelity (1-5)**
57  Evaluate both the logical coherence of the thinking process AND whether it stays completely faithful to the
    original audio description without introducing fabricated details.
58
59  SCORING GUIDE:
60  - **Score 5 (Excellent)**: Perfect logical flow with clear, sound reasoning that seamlessly connects the
    question, audio analysis, and correct answer. The thinking process is highly coherent, well-structured, AND
    stays completely faithful to the original audio description with no additional details or fabricated content
    introduced. Perfect adherence to source material with excellent logic.
61  - **Score 4 (Good)**: Strong logical reasoning with clear connections between components, though may have
    minor gaps or slightly unclear transitions. The thinking process is mostly faithful to the audio description
    with only very minor, well-supported inferences. Good logic with minimal risk of hallucination.
62  - **Score 3 (Average)**: Generally logical reasoning with some clear connections, but may have noticeable gaps
     or unclear steps in the reasoning process. The thinking process includes some reasonable inferences or
    interpretations that stay generally within the bounds of the audio description. Acceptable logic with some
    minor speculation.
63  - **Score 2 (Poor)**: Weak logical flow with significant gaps, unclear reasoning, or poor connections between
    the question, audio analysis, and answer. OR the thinking process introduces several details or facts not
```

present in the audio description, or makes significant assumptions not supported by the source material. Poor logic or notable hallucination issues.
64  - **Score 1 (Very Poor)**: Illogical or incoherent reasoning with major flaws, contradictions, or complete failure to properly connect the components. OR the thinking process contains extensive fabricated details, imagined content, or information that directly contradicts or goes far beyond what's in the audio description. Severe logic problems or hallucination issues.
65
66  **ASPECT 5: Simple Thinking Process Logic & Fidelity (1-5)**
67  Evaluate both the logical coherence of the simple thinking process AND whether it stays completely faithful to the original audio description without introducing fabricated details.
68
69  SCORING GUIDE:
70  - **Score 5 (Excellent)**: Perfect logical flow with clear, sound reasoning that seamlessly connects the question, audio analysis, and correct answer. The simple thinking process is highly coherent, well-structured, AND stays completely faithful to the original audio description with no additional details or fabricated content introduced. Perfect adherence to source material with excellent logic.
71  - **Score 4 (Good)**: Strong logical reasoning with clear connections between components, though may have minor gaps or slightly unclear transitions. The simple thinking process is mostly faithful to the audio description with only very minor, well-supported inferences. Good logic with minimal risk of hallucination.
72  - **Score 3 (Average)**: Generally logical reasoning with some clear connections, but may have noticeable gaps or unclear steps in the reasoning process. The simple thinking process includes some reasonable inferences or interpretations that stay generally within the bounds of the audio description. Acceptable logic with some minor speculation.
73  - **Score 2 (Poor)**: Weak logical flow with significant gaps, unclear reasoning, or poor connections between the question, audio analysis, and answer. OR the simple thinking process introduces several details or facts not present in the audio description, or makes significant assumptions not supported by the source material. Poor logic or notable hallucination issues.
74  - **Score 1 (Very Poor)**: Illogical or incoherent reasoning with major flaws, contradictions, or complete failure to properly connect the components. OR the simple thinking process contains extensive fabricated details, imagined content, or information that directly contradicts or goes far beyond what's in the audio description. Severe logic problems or hallucination issues.
75
76  OUTPUT FORMAT:
77  You must provide exactly five scores in the following format:
78  <aspect1_score>X</aspect1_score>
79  <aspect2_score>Y</aspect2_score>
80  <aspect3_score>Z</aspect3_score>
81  <aspect4_score>W</aspect4_score>
82  <aspect5_score>V</aspect5_score>
83
84  Where X, Y, Z, W, V are integers from 1 to 5.
85
86  IMPORTANT:
87  - Each score must be a single integer between 1 and 5
88  - Do not include any explanations or additional text outside the score tags
89  - Focus on objective evaluation based on the criteria provided

## B.6 MODEL EVALUATION PROMPTS

```
1  # Audio-Flamingo 2
2  [Question] (A) Option1. (B) Option2. (C) Option3. (D) Option4.
3
4  # R1-AQA
5  [Question] Please choose the answer from the following options: ['Option1', 'Option2', 'Option3', 'Option4'].
   Output the final answer in <answer> </answer>.
6
7  # Kimi-Audio and Audio-Reasoner
8  [Question] A. Option1 B. Option2 C. Option3 D. Option4
```

## B.7 MODEL TRAINING PROMPTS (QWEN2.5-OMNI)

```
1  # All Data SFT
2  [question] Please choose the answer from the following options: A. Option1 B. Option2 C. Option3 D. Option4
3
4  # Others
5  [Question] Please choose the answer from the following options: ['Option1', 'Option2', 'Option3', 'Option4'].
   Output the final answer in <answer> </answer>.
6
7  # System Prompt
8  You are an audio understanding model that answers multiple choice questions based on audio content.
```

## C  QUALITY VERIFICATION WITHIN THE PIPELINE

The AudioMCQ construction pipeline incorporates a fine-grained, multi-step quality control process to ensure data integrity:

1. **Automated Validation:** All generated data points must pass a custom format checker for immediate structural compliance.

2. **Iterative LLM Correction:** Data failing the initial format check is automatically sent back to the Large Language Model for re-generation until the required format is met.

3. **Manual Spot-Checking:** Batches of data undergo quality assurance via manual review by a team of $2-3$ human annotators.

4. **System Refinement:** Issues identified during manual review necessitate modifications to the LLM prompting strategy and checker rules, followed by a full re-generation of the corresponding dataset stage to implement systemic corrections.

## D  EXAMPLES OF TWO CASES OF ZERO AUDIO-CONTRIBUTION

### D.1  EXPLICIT LOGICAL REASONING

**Q:** What mechanical sound is continuously present in the background?

**A.** The revving motorcycle
**B.** The running engine (correct)
**C.** The passing train
**D.** The screeching brakes

**Qwen3-235B Analysis:** The question asks: "What mechanical sound is continuously present in the background?" The key word in the question is "continuously," which implies a sustained, ongoing sound rather than an intermittent or brief one. Now, examine the answer choices in light of this clue: "The revving motorcycle" – Revving is typically intermittent, involving bursts of sound, not continuous. "The running engine" – An engine that is running produces a steady, continuous sound, matching the description of "continuously present." "The passing train" – A train passing by produces a sound that increases and then decreases in volume; it is temporary and not continuous. "The screeching brakes" – Screeching brakes are usually short-lived and abrupt, not continuous. Based solely on the textual clues—especially the word "continuously"—we can logically infer that "The running engine" is the only option that describes a mechanical sound that would be consistently present in the background. The other options describe transient or momentary sounds. Therefore, the correct answer can be determined from the text alone, without needing to hear the audio.

**Q:** What specific sound do the car's tires make as it accelerates?

**A.** They squeal slightly while moving fast (correct)
**B.** They screech loudly while stopping
**C.** They crunch over gravel rapidly
**D.** They hum quietly on the highway

**Qwen3-235B Analysis:** The question asks about the specific sound the car's tires make as it accelerates. The correct answer is "They squeal slightly while moving fast." Looking at the answer choices, each describes a different tire sound associated with a specific driving condition: "They squeal slightly while moving fast" implies acceleration with some tire grip stress; "They screech loudly while stopping" refers to braking, not accelerating; "They crunch over gravel rapidly" suggests driving on a gravel surface; "They hum quietly on the highway" indicates steady-speed driving. The key clue in the question is "as it accelerates," which directly relates to the moment the car increases speed. Among the options, only "squeal slightly while moving fast" plausibly corresponds to tire sound during acceleration, as tire squeal often occurs due to increased traction demand when speeding up. The other choices are tied to different actions (braking, driving on gravel, cruising), which are inconsistent with acceleration. Therefore, using only the textual clues — particularly "accelerates" — and matching it to the most contextually appropriate sound from the choices, one can deduce the correct answer without hearing the audio.

**Q:** What type of vehicle can be heard accelerating quickly?

**A.** A car starting slowly
**B.** A train approaching
**C.** A bicycle moving quietly

**D.** A motorcycle speeding away (correct)

**Qwen3-235B Analysis:** The question asks: "What type of vehicle can be heard accelerating quickly?" The key clue in the text is the phrase "accelerating quickly," which implies a vehicle that produces a noticeable and rapid increase in speed, typically associated with a loud or sudden engine sound. Now, examining the answer choices: "A car starting slowly" – contradicts "accelerating quickly" due to "slowly." "A train approaching" – while trains can be loud, "approaching" does not necessarily imply acceleration, and trains are less commonly described as "accelerating quickly" in everyday auditory contexts. "A bicycle moving quietly" – "quietly" contradicts the idea of being clearly heard accelerating; also, bicycles do not have engines and typically don't produce loud acceleration sounds. "A motorcycle speeding away" – "speeding away" directly aligns with "accelerating quickly," and motorcycles are commonly associated with rapid acceleration and loud engine noises. All necessary information to eliminate incorrect choices and select the correct one is present in the textual description. The phrase "accelerating quickly" provides a strong textual clue that, when matched with the descriptors in the answer choices, allows identification of "A motorcycle speeding away" as the most logically consistent option–without needing to hear the audio.

---

**Q:** What mechanical auditory pattern occurs continuously throughout the audio?

**A.** A siren fluctuating in pitch periodically
**B.** A machine starting and stopping abruptly
**C.** The sound of wind increasing and decreasing
**D.** An engine maintaining steady operation (correct)

**Qwen3-235B Analysis:** The question asks: "What mechanical auditory pattern occurs continuously throughout the audio?" The key word in the question is "continuously," which implies that the correct answer must describe a sound that does not stop or fluctuate significantly—it must be unbroken and steady over time. Now, examine the answer choices in light of the word "continuously": "A siren fluctuating in pitch periodically" – The word "periodically" indicates a repeating change, meaning the sound is not steady or continuous in pitch; it varies. This contradicts a truly continuous pattern. "A machine starting and stopping abruptly" – "Starting and stopping" clearly indicates an intermittent pattern, not a continuous one. "The sound of wind increasing and decreasing" – "Increasing and decreasing" suggests variation over time, not a steady, continuous state. "An engine maintaining steady operation" – "Maintaining steady operation" directly implies a consistent, unchanging, and continuous mechanical sound. Given that the question explicitly uses the word "continuously," and only one choice describes a truly continuous and unchanging mechanical auditory pattern, the correct answer can be deduced from the textual clues alone, without needing to hear the audio.

---

**Q:** What mechanical sound remains constant throughout the audio?

**A.** A car horn beeping
**B.** An engine running steadily (correct)
**C.** A motor turning on and off
**D.** A siren approaching quickly

**Qwen3-235B Analysis:** The question asks: "What mechanical sound remains constant throughout the audio?" The key phrase in the question is "remains constant throughout the audio." This implies that the correct answer must describe a sound that is unchanging or continuous in nature. Now, evaluating the answer choices based on this clue: "A car horn beeping" – This is intermittent, not constant. "An engine running steadily" – The word "steadily" directly indicates a constant, unchanging sound. "A motor turning on and off" – This is cyclical and not constant. "A siren approaching quickly" – This involves a changing pitch (Doppler effect) and is not constant. Even without hearing the audio, the phrase "remains constant" allows us to logically deduce that "An engine running steadily" is the only option that matches the description of a continuous, unchanging mechanical sound. The textual clues in the question and the semantics of the answer choices are sufficient to identify the correct answer.

### D.2 Implicit Knowledge Retrieval

**Q:** What specific actions are clearly heard in the audio?

**A.** Ducks quacking and liquid splashing (correct)
**B.** Birds chirping and wind blowing
**C.** Ducks flying and thunder rumbling
**D.** Frogs croaking and water dripping

**Qwen3-235B Analysis:** The question asks: "What specific actions are clearly heard in the audio?" This implies that the answer depends on auditory content — specifically, which sounds are perceptible in the audio. The answer choices are all descriptions of soundscapes (e.g., "ducks quacking and liquid splashing", "birds chirping and wind blowing"), and the correct answer is stated to be "Ducks quacking and liquid

splashing". However, the task is to determine whether this question can be answered **based solely on textual clues in the question**, without listening to the audio. The question itself does not contain any explicit textual indicators about what is heard — there are no descriptions of the sounds, no metaphors, no contextual hints (such as a setting like "a pond during a storm") that would allow a reader to infer which sounds are present. All the answer choices are plausible in certain environments, but without additional descriptive context in the question text, there is no way to determine which one is correct based on text alone. The identification of the correct answer relies entirely on hearing the audio or having a textual description of its contents, which is not provided. Therefore, the correct answer cannot be determined from the text alone.

---

**Q:** What is the sequence of audio elements at the beginning?

**A.** Steam hissing followed by speech (correct)
**B.** Music playing followed by speech
**C.** Footsteps approaching followed by speech
**D.** Wind blowing followed by speech

**Qwen3-235B Analysis:** The question asks about the "sequence of audio elements at the beginning," which inherently refers to the temporal order of sounds in an audio clip. The answer choices describe specific sound events (e.g., "steam hissing," "music playing") followed by speech, which are auditory in nature. The question text itself does not provide any explicit description of the audio content, such as "the audio starts with steam hissing" or any equivalent textual clue. Without such descriptive information embedded in the question text, it is impossible to determine the correct sequence of audio elements based solely on the text. Therefore, one must rely on listening to the audio to identify which sound comes first and whether it is followed by speech. Since the textual clues in the question are insufficient to determine the correct answer, the audio content is necessary.

---

**Q:** What occurs while the female speaker is continuously talking?

**A.** A door slams shut loudly
**B.** A spraying sound happens several times (correct)
**C.** A dog barks intermittently
**D.** A phone rings in the distance

**Qwen3-235B Analysis:** The question asks: "What occurs while the female speaker is continuously talking?" This implies that the correct answer must describe an event that happens simultaneously with the female speaker's continuous speech. The correct answer is "A spraying sound happens several times." To determine whether this can be answered using only the textual clues, we must assess if the question itself contains enough information to deduce the correct answer without hearing the audio. However, the question text does not mention any details about a spraying sound, a door, a dog, or a phone. There are no descriptive hints, onomatopoeia, or contextual clues (e.g., setting, activity, or associated actions like gardening or a phone call) that would allow one to infer that a spraying sound occurs. All answer choices are auditory events that could plausibly occur in the background, but nothing in the text favors one over the others based on logic or implication. Therefore, without the audio, it is impossible to determine which sound occurs during the female speaker's continuous talking. The text of the question alone does not provide sufficient clues to arrive at the correct answer.

---

**Q:** What specific sound event occurs in the audio?

**A.** A motorcycle increases speed moving away (correct)
**B.** A car engine idling steadily
**C.** A motorcycle decreases speed approaching
**D.** A bicycle speeding on pavement

**Qwen3-235B Analysis:** The question asks: "What specific sound event occurs in the audio?" and provides four answer choices describing different sound events involving vehicles. The correct answer is stated to be "A motorcycle increases speed moving away." However, the question text itself does not contain any descriptive clues about what is happening in the audio—there are no onomatopoeic words, contextual hints, or narrative descriptions that would allow one to infer the sound event. The identification of the correct answer depends entirely on interpreting the auditory characteristics (such as the Doppler effect, engine pitch changes, etc.) present in the audio, which are not described or hinted at in the text. Since the textual information provided in the question does not include any clues to determine the sound event without listening to the audio, the correct answer cannot be deduced from the text alone.

---

**Q:** What combination of natural and animal sounds is present in the audio?

**A.** Wind blowing with insect buzzing and bird chirping (correct)

---

**B.** Ocean waves with seagull crying and boat horn
**C.** Leaves rustling with dog barking and footsteps
**D.** Rain falling with frog croaking and thunder rumbling

**Qwen3-235B Analysis:** The question asks about the specific combination of natural and animal sounds present in an audio file. The answer choices provide different combinations of environmental and animal sounds, such as wind, insects, birds, ocean waves, seagulls, frogs, etc. However, the question text itself does not include any explicit description or textual clue about what sounds are actually in the audio. It only frames the query and offers multiple choices. There is no additional context, transcript, or onomatopoeic hints within the question text that would allow one to deduce the correct combination without hearing the audio. Therefore, it is not possible to determine the correct answer using only the textual clues in the question. Listening to the audio is necessary to identify the actual sounds present.

# E EXPERIMENT DETAILS

## E.1 HYPERPARAMETER CONFIGURATION

We present the detailed hyperparameters used in our experiments. All experiments were conducted using the same hardware setup to ensure fair comparison.

Table 6: Hyperparameters for SFT.

| Parameter | Value |
|---|---|
| Training Type | Full Parameter |
| Data Type | bfloat16 |
| Max Sequence Length | 1024 |
| Training Epochs | 2 |
| Batch Size (per device) | 20 |
| Learning Rate | 1e-6 |
| Gradient Accumulation Steps | 1 |
| Warmup Ratio | 0.05 |
| LR Scheduler | Cosine |
| Optimization | DeepSpeed ZeRO-2 |

Table 7: Hyperparameters for GRPO.

| Parameter | Value |
|---|---|
| Training Type | Full Parameter |
| Data Type | bfloat16 |
| Max Completion Length | 1024 |
| Training Steps | 1000 |
| Batch Size (per device) | 8 |
| Learning Rate | 1e-6 |
| Warmup Ratio | 0.05 |
| Number of Generations | 8 |
| Temperature | 1.5 |
| Top-k Sampling | 4 |
| KL Divergence Penalty ($\beta$) | 0.001 |
| LR Scheduler | Cosine |
| Optimization | DeepSpeed ZeRO-2 |

## E.2 EXPERIMENTAL CONTROLS FOR SFT-TO-RL PIPELINE

To ensure the reliability and validity of our experimental results, we implement strict controls across multiple dimensions:

- **Fixed Data Volume:** The SFT training data volume is fixed at 313,177 samples, corresponding exactly to the low audio-contribution split size in AudioMCQ, ensuring consistent comparison across different experimental conditions.

- **Data Augmentation:** During training, each multiple-choice question is replicated four times with randomized option orders to reduce position bias.

- **Data Isolation:** SFT and RL training datasets are kept strictly non-overlapping to prevent data leakage and ensure that performance improvements can be attributed solely to the RL training methodology.

- **SFT Model Selection:** During SFT, we evaluate checkpoints at three intermediate steps (750, 1000, and 1250) and select the best-performing model based on MMAU-test-mini-4k evaluation scores to mitigate training variance effects.

- **RL Training Protocol:** The RL phase consists of exactly 1000 training steps across all experiments, with final checkpoint selection based on comprehensive performance evaluation on MMAU-test-mini-4k.

### E.3 PERFORMANCE OF OUR METHODS

Table 8: Detailed performance of our methods. Model A: All Data SFT; Model B: All Data GRPO; Model C: Mix AC SFT + Mix AC GRPO; Model D: Weak AC SFT + Strong AC GRPO; Model E: Mix AC SFT + Strong AC GRPO.

| MMAU-test-mini Performance by Subset (%) | | | | | | | |
|---|---|---|---|---|---|---|---|
| Subset | Model A | Model B | Model C | Model D | Model E | Average | Random Guess |
| Sound | 79.6 | 82.6 | 76.0 | 83.8 | 79.3 | **80.3** | 25.0 |
| Music | 72.5 | 74.9 | 71.6 | 72.2 | 72.8 | **72.8** | 25.0 |
| Speech | 73.6 | 76.9 | 75.1 | 78.7 | 77.2 | **76.3** | 26.7 |
| **Overall** | **75.2** | **78.1** | **74.2** | **78.2** | **76.4** | **76.4** | **25.5** |
| MMAU Performance by Subset (%) | | | | | | | |
| Subset | Model A | Model B | Model C | Model D | Model E | Average | Random Guess |
| Sound | 78.2 | 80.3 | 76.4 | 79.0 | 78.6 | **78.5** | - |
| Music | 72.2 | 68.8 | 70.6 | 71.7 | 70.4 | **70.7** | - |
| Speech | 74.6 | 77.0 | 76.1 | 76.3 | 76.5 | **76.1** | - |
| **Overall** | **75.0** | **75.4** | **74.4** | **75.6** | **75.1** | **75.1** | **-** |
| MMAR Performance by Subset (%) | | | | | | | |
| Subset | Model A | Model B | Model C | Model D | Model E | Average | Random Guess |
| Perception | 61.4 | 64.1 | 64.4 | 64.1 | 68.6 | **64.5** | 27.2 |
| Semantic | 67.5 | 64.3 | 66.8 | 67.5 | 67.0 | **66.6** | 31.4 |
| Signal | 67.4 | 48.8 | 55.8 | 60.5 | 60.5 | **58.6** | 33.0 |
| Cultural | 64.5 | 60.3 | 63.8 | 63.8 | 64.5 | **63.4** | 28.4 |
| **Overall** | **64.6** | **63.0** | **64.9** | **65.3** | **67.0** | **65.0** | **29.3** |
| MMSU Performance by Subset (%) | | | | | | | |
| Subset | Model A | Model B | Model C | Model D | Model E | Average | Random Guess |
| Perception | 49.8 | 60.4 | 61.7 | 60.1 | 64.2 | **59.2** | 25.0 |
| Reasoning | 79.2 | 80.7 | 77.2 | 79.3 | 79.8 | **79.2** | 25.0 |
| **Overall** | **64.0** | **70.2** | **69.2** | **69.3** | **71.7** | **68.9** | **25.0** |

