# OpenReview forum: "Measuring Audio's Impact on Correctness: Audio-Contribution-Aware Post-Training of Large Audio Language Models"
_ICLR.cc/2026/Conference — ICLR 2026 Poster_

### Official Review · Reviewer_dnN2 · 2025-10-30

[review text omitted: it was posted to a different submission]

---

> ### Author Response · Authors · 2025-11-14
> **Author Response to Reviewer Comments (to Weaknesses)**
>
> Hello,
>
> Thank you very much for your detailed and rigorous assessment of our manuscript. We would like to politely clarify a few points regarding the novelty of our methods and the scope of our evaluation.
>
> ## Response to Weaknesses
>
> ### 1. Weak-to-strong strategies lack novelty.
>
> We respectfully point out that, to our knowledge, our paper introduces the **first** data allocation methodology—**Audio Contribution Filtering**—specifically tailored for post-training Large Audio-Language Models (LALMs).
>
> Consequently, the weak-to-strong and mixed-to-strong strategies combined with SFT and GRPO, when **based on this specific allocation method**, are novel in the LALM domain. We also list some recent LALM post-training frameworks:
>
> * **Pure GRPO:** (e.g., R1-AQA [[https://arxiv.org/abs/2503.11197](https://arxiv.org/abs/2503.11197)], Omni-R1 [[https://arxiv.org/abs/2505.09439](https://arxiv.org/abs/2505.09439)])
> * **Curriculum Learning:** Based solely on difficulty, not on the distinct audio contribution component (e.g., SARI [[https://arxiv.org/abs/2504.15900](https://arxiv.org/abs/2504.15900)])
>
> ### 2. Experimental evaluation is not comprehensive.
>
> We believe our experimental evaluation is comprehensive within the scope of current LALM post-training literature. We utilize a combination of four benchmarks: **full MMAU, MMAU-test-mini, MMAR, and MMSU**.
>
> These are the standard, established metrics for LALM post-training. For instance, related works typically use a subset:
>
> * R1-AQA [[https://arxiv.org/abs/2503.11197](https://arxiv.org/abs/2503.11197)]: MMAU (test-mini only)
> * Omni-R1 [[https://arxiv.org/abs/2505.09439](https://arxiv.org/abs/2505.09439)]: MMAU, MMAR
> * SARI [[https://arxiv.org/abs/2504.15900](https://arxiv.org/abs/2504.15900)]: MMAU (test-mini only), MMSU
>
> By incorporating all four, our evaluation system is one of the most complete and rigorous benchmarks currently applied for LALM post-training.
>
> ### 3. Redundant analysis and lack of concrete novelty.
>
> We appreciate your critique regarding the paper's writing and structural issues. We aimed to provide detailed transparency into our findings and the complex interplay of audio components, which may have led to some redundancy. We sincerely apologize for any confusion caused by the current expression of our analysis. We would be grateful if you could provide specific examples of the confusing or redundant analysis so we may address them most effectively.
>
> ***

---

> ### Author Response · Authors · 2025-11-14
> **Author Response to Reviewer Comments (to Questions)**
>
> ## Response to Questions
>
> ### 1. Dataset utilization, well-defined evidence, and public availability.
>
> We apologize for not clarifying the release status in the initial submission. We confirm that the **AudioMCQ dataset and the corresponding code will be made publicly available very soon**. Since it has not been released, it is not utilized by other algorithms yet.
>
> The evidence that the dataset is "well-defined and meaningful" is demonstrated empirically in Section 4.1 (DATASET QUALITY VALIDATION), where we show that models trained on **AudioMCQ achieve superior performance** with fewer GRPO steps (1,200 steps) compared to models trained on alternative, state-of-the-art LALM datasets (1,700 steps).
>
> ### 2. Human annotation and quality verification.
>
> Sorry for the confusion we made. Yes, human annotation and verification are integral to the AudioMCQ construction pipeline. We employed a fine-grained, multi-step quality control process for every step of data generation (referencing Figure 1):
>
> 1.  **Automated Validation:** Every generated data point passes a custom-written format checker for immediate structural validation.
> 2.  **Iterative LLM Correction:** If the format fails, the data is sent back to the LLM for re-generation until correctness is achieved.
> 3.  **Manual Spot-Checking:** Batches undergo manual review by a team of 2-3 annotators.
> 4.  **Prompt Refinement:** Any issues identified during manual review trigger modifications to the LLM prompting strategy and the checker rules, followed by a full re-generation of that entire dataset stage.
>
> The detailed prompts used for generation are provided in Appendix B. **Additionally, we have included a new section, Appendix C: QUALITY VERIFICATION WITHIN THE PIPELINE, for your review.**
>
> ### 3. Potential research impact.
>
> The potential research impact of AudioMCQ and our accompanying analysis is substantial:
>
> * **Dedicated Resource for Post-Training:** AudioMCQ is one of the first and largest datasets explicitly designed for the post-training and fine-tuning of LALMs (unlike existing benchmarks which may be vision-centric).
> * **Methodological Insight (Zero-Audio-Contribution):** Our experiments rigorously uncover and study the prevalent "zero-audio-contribution" phenomenon in current LALM benchmarks. This finding, echoing similar recent work in the vision domain (e.g., [https://arxiv.org/abs/2511.04655](https://arxiv.org/abs/2511.04655)), is crucial for compelling researchers to address the fundamental question: **"Is the LALM truly listening to the audio?"** This will drive the design of more robust benchmarks and LALM architectures.
> * **Theoretical Framework for SFT-RL:** Our study provides a principled guideline for the combined SFT-RL post-training framework. We demonstrate that random data allocation can cause the SFT-RL framework to underperform compared to pure RL. Our **Audio Contribution Filtering** method offers a theoretically grounded way to **safely and efficiently leverage the entirety of a large post-training dataset** within the SFT-RL pipeline, which is essential as data scales up.
>
> We hope these detailed clarifications address your concerns and demonstrate the novelty of our data allocation method and the results achieved through our evaluation.

---

### Official Review · Reviewer_DR3o · 2025-10-31

**Soundness:** 3
**Presentation:** 3
**Contribution:** 3
**Rating:** 8
**Confidence:** 4

**Summary:**

This paper investigates whether audio contributes to answering audio-based questions and the authors create an audio Q&A dataset and discover that models often answer correctly using just the text context, ignoring the audio. They split the training data into cases where audio is actually needed vs. not, and then train the model in stages, focusing on the audio-critical examples. This targeted training leads to better performance on audio question answering tasks.

**Strengths:**

- The paper introduces AudioMCQ, a large-scale audio multiple-choice question dataset with 571,000+ samples.
- Identification of zero audio-contribution issues: The authors systematically measure how often models answer correctly without using the audio. By replacing audio, they find that current LALMs still achieve far-above-chance accuracy.
- They propose ACF to split training data based on whether the audio actually affects correctness, with data partitioning, allowing the training to focus differently on audio-easy vs audio-critical examples.

**Weaknesses:**

- All experiments and the dataset use a multiple-choice Q&A setup. The method’s effectiveness on open-ended or free-form audio questions remains untested.
- The ACF procedure uses predictions from three existing LALMs to decide if audio is needed for each sample, this automatic labeling might be noisy or biased.
- The two training paradigms showed different strengths across benchmarks, and no single approach uniformly excels. For example, the Weak-to-Strong method achieved the best results on the MMAU benchmark but underperformed on datasets where audio cues were crucial, where Mixed-to-Strong worked better. This indicates the training strategy must be aligned to the task’s audio-contribution type, which further adds complexity.

**Questions:**

- How can the audio-contribution-aware approach be applied to more open-ended or dialog-style audio question answering tasks that don’t have predefined options?
- The dataset provides detailed CoT reasoning for each question. However, the paper does not isolate how much these reasoning prompts actually improve model performance or reasoning ability. Would the LALM perform significantly worse without the CoT training data, or do the benefits mainly come from the RL fine-tuning and audio-centric data filtering?

---

> ### Author Response · Authors · 2025-11-19
> **Response to Reviewer DR3o (Answering Weakness 1 & Question 1: Open-ended Audio QA + Weakness 2: Robustness of ACF Voting)**
>
> We express our **sincere gratitude** for your positive assessment and your recognition of the significance of AudioMCQ and the audio-contribution issue. We value your insightful feedback regarding the extension to open-ended tasks and the robustness of our data partitioning. Below, we address your concerns point-by-point. **All modified content has been marked in blue in the revised PDF.**
>
> ### **1. Response to Weakness 1 & Question 1: Open-ended Audio QA**
>
> **Reviewer Comment:** Concerns about the limitation to multiple-choice formats and how the method applies to open-ended or dialogue-style tasks.
>
> **Response:**
> We appreciate this forward-looking question. Extending Audio-Contribution Filtering (ACF) to open-ended generation is indeed a critical direction. In fact, this topic constitutes a central component of our **next major publication**. We are happy to offer a brief, preliminary insight into how we approach open-ended tasks:
>
> Formally, let $\mathbf{a} = \{a_1, \dots, a_M\}$ be the sequence of input audio tokens and $\mathbf{q} = \{q_1, \dots, q_N\}$ be the sequence of question text tokens. While a standard multimodal model generates the target response $\mathbf{y} = \{y_1, \dots, y_T\}$ conditioned on the joint input $[\mathbf{a}; \mathbf{q}]$, we aim to isolate the extent to which the model relies purely on textual cues.
>
> To achieve this, we evaluate the likelihood of generating the ground-truth response $\mathbf{y}$ conditioned exclusively on the text question $\mathbf{q}$. We measure this using the **Perplexity (PPL)**, normalized by the response length $T$:
>
> $$
> \text{PPL}(\mathbf{y} \mid \mathbf{q}) = \exp\left( -\frac{1}{T} \sum_{t=1}^{T} \log P_\theta(y_t \mid \mathbf{q}, y_{<t}) \right)
> $$
>
> where $P_\theta$ denotes the model's probability distribution, and $y_{<t}$ represents the autoregressive history of response tokens.
>
> We subsequently define the **Weak Audio-Contribution Score** ($\mathcal{S}_{\text{weak}}$) as the reciprocal of this text-only perplexity:
>
> $$
> \mathcal{S}_{\text{weak}} = \frac{1}{\text{PPL}(\mathbf{y} \mid \mathbf{q})}
> $$
>
> Conceptually, a higher $\mathcal{S}_{\text{weak}}$ corresponds to a lower perplexity, indicating that the response is highly predictable given the text alone. This suggests that the model's reliance on the audio modality is minimal (or "weak") for correctly inferring the answer.
>
> **Action:** To **preserve the innovative edge** of our forthcoming publication, we respectfully request that these details remain **excluded from the current article**. However, we agree that this is the logical next step.
>
> ### **2. Response to Weakness 2: Robustness of ACF Voting**
>
> **Reviewer Comment:** Concerns that the 3-model voting mechanism might be noisy or biased.
>
> **Response:**
> We acknowledge the concern regarding the noise in automated labeling.
>
> We selected the voting models (R1-AQA, Audio-Flamingo2, Kimi-Audio) based on the state-of-the-art landscape available **during our data construction phase (April 2025).** At that time, we did not have many models to choose from (Qwen-audio, Qwen2-audio, R1-AQA, Audio-Flamingo, Audio-Flamingo2, Kimi-Audio). To minimize bias and ensure fairness, we can only select three models from **distinct families where possible**, specifically avoiding overlap within families (e.g., grouping Qwen-Audio and Qwen2-Audio).
>
> We believe the most significant proof of the partition's validity lies in the downstream performance. As shown in **Appendix E.3, Table 8**, models trained on our partitioned data (based on weak/strong contribution) **show significant improvements in perception metrics.** This performance gain serves as empirical validation that the current voting-based split is effective in distinguishing audio-critical samples.
>
> **Action:** We have updated the **Conclusion** to reflect the need for continued improvement in partitioning robustness: *"Future work could focus on establishing a **more robust audio-contribution-based data partitioning for AudioMCQ**, extending Audio-Contribution Filtering to general audio question-answering tasks, and further investigating CoT reasoning capabilities in audio understanding contexts."*

---

> > ### Author Response · Authors · 2025-11-19
> > **Response to Reviewer DR3o (Answering Weakness 3: Training Strategy Trade-offs and Question 2: Chain-of-Thought (CoT))**
> >
> > ### **3. Response to Weakness 3: Training Strategy Trade-offs**
> >
> > **Reviewer Comment:** Noted that Weak-to-Strong and Mixed-to-Strong have different strengths (e.g., on MMAU vs. MMAR), suggesting complexity in aligning strategy to task.
> >
> > **Response:**
> > We respectfully **maintain that our paradigm instills generalizable robustness**, even if specific benchmarks favor different SFT strategies.
> >
> > 1.  **Universal Perception Gain:** Crucially, both **Weak-to-Strong** and **Mixed-to-Strong** significantly outperform the Mixed-to-Mixed baseline. For instance, Weak-to-Strong achieves the best results on MMAU while still surpassing baselines on strong-audio benchmarks like MMAR. This indicates that the **GRPO stage using strong audio-contribution data** provides a universal enhancement to the model's fundamental "listening" capability.
> > 2.  **Role of SFT:** We view the SFT data selection as a downstream **engineering choice** to align the model with specific task formats (or benchmark distributions), **under the premise** that the fundamental perception capabilities have already been secured by our Strong-Audio GRPO.
> >
> > **Action:** We have clarified this distinction in the final paragraph of **Section 4.2**:
> > *"In contrast, GRPO with strong audio-contribution data yields significant improvements across all benchmarks and their AC-strong splits, reflecting a substantial gain in the model's perception capability, particularly pronounced on the strong audio-contribution benchmarks (MMAR and MMSU). Second, SFT data selection should align with specific downstream task characteristics."*
> >
> > We wish to further clarify our perspective: the issue lies not in the generalizability of our post-training method, but in the fact that **current desgins of models and benchmarks overlook the critical metric of 'audio-contribution,'** resulting in significant inconsistency across benchmarks. We firmly believe that Large Audio Language Models **must reason strictly based on audio input rather than relying on textual priors or guessing**, as the latter severely compromises practical utility. Similarly, LALM benchmarks must inherently demand **strong audio-contribution.** Therefore, we consider our strong audio-contribution GRPO to be more pivotal than SFT. GRPO aligns the model with the correct modality dependence, whereas SFT—as previously noted—remains merely a pragmatic engineering expedient in the current landscape.
> >
> > ***
> > **Revision: Refined Checkpoint Selection for Mixed-to-Strong Strategy**
> >
> > Following a comprehensive retrospective analysis of the **Mixed-to-Strong** training trajectory, we identified a superior checkpoint at **500 GRPO steps** (succeeding the initial 1000 mixed-SFT steps). While **getting the same score** with the previously selected 900-step checkpoint on the validation set, this earlier model demonstrates stronger generalization across primary benchmarks.
> >
> > The comparative results are summarized below:
> >
> > | Benchmark | **New 500-step Checkpoint** | Previous 900-step Checkpoint |
> > | :--- | :---: | :---: |
> > | **MMAU-test-mini** | $\mathbf{76.4}$ | $75.8$ |
> > | **MMAU** | $\mathbf{75.1}$ | $74.6$ |
> > | **MMAR** | $67.0$ | $\mathbf{67.1}$ |
> > | **MMSU** | $\mathbf{71.7}$ | $70.7$ |
> >
> > **Action:** We have updated **Table 2** and the corresponding analysis in the manuscript to reflect the performance of this improved 500-step model. We have added a **Figure 4** showing the selection of the best checkpoint during the experiments.
> > ***
> >
> > ### **4. Response to Question 2: Chain-of-Thought (CoT)**
> >
> > **Reviewer Comment:** Questions whether CoT data improves performance or if benefits come mainly from RL/filtering, noting the paper doesn't isolate CoT effects.
> >
> > **Response:**
> > We apologize for any confusion. To clarify: **We did not use CoT in the training experiments presented in this paper.** While we constructed CoT annotations for the dataset to support future research, our current evaluation focuses strictly on **perceptual fidelity** rather than reasoning. The performance gains reported are derived entirely from our **Audio-Contribution Filtering and the Strong-Audio GRPO**, without any CoT supervision. This further highlights the effectiveness of our proposed data partitioning strategy.
> >
> > Furthermore, we are **actively conducting a series of experiments focused on CoT**, which will constitute the core of **our upcoming publication**. We would like to share a preliminary insight from this ongoing work: We demonstrate that CoT improves reasoning in audio understanding. By integrating 'strong audio-contribution' data and other methods, we aim to **synergize this reasoning capability with robust fundamental perception.**
> >
> > **Action:** We have updated the first paragraph of **Section 4 Experiments** to explicitly state this exclusion: *"Given our focus on perceptual fidelity rather than reasoning, CoT is excluded from the following evaluations."*
> >
> > Thank you again for your strong endorsement of our work!

---

### Official Review · Reviewer_YhaS · 2025-10-31

**Soundness:** 3
**Presentation:** 3
**Contribution:** 4
**Rating:** 6
**Confidence:** 4

**Summary:**

The paper presents AudioMCQ, a comprehensive audio multiple-choice question dataset comprising 571k samples with two kinds of chain-of-thought annotations and investigates the prevalent zero audio-contribution phenomenon in LALMs, where models derive correct answers solely from textual information without processing audio content. To address this, the authors propose an Audio-Contribution Filtering method to partition data into 'weak' and 'strong' subsets. Based on this, they introduce effective post-training paradigms, such as Weak-to-Strong, which achieve new state-of-the-art performance on benchmarks like MMAU, MMAR, and MMSU.

**Strengths:**

* The paper is well written.
* The dataset collection process is very thoroughly explained.
* The quantification of this "zero audio-contribution" scenario (e.g., 49.8% average accuracy on MMAU-test-mini with silent audio) is very relevant to the area.
* The experiment setup is complete and shows the benefits of the proposed post-training paradigms.
* The paper achieves new and significant state-of-the-art results on several diverse and challenging audio understanding benchmarks (MMAU, MMAR, MMSU).

**Weaknesses:**

1. There are no experiments with the CoT data collected from the dataset. These experiments would not only "further investigate the CoT reasoning capabilities" of models but actually solidify the contribution of the dataset in this end. Without this experiment, it is hard to evaluate if these CoT traces' quality.
2. The entire Audio-Contribution Filtering (ACF) method hinges on a majority vote from three specific LALMs. This choice is not justified, ablated, or analyzed for sensitivity. It is unclear how robust the resulting "weak" and "strong" data partitions are. A different set of filtering models could potentially lead to a different partition, which would call into question the generality of the training paradigms built upon them.
3. The AudioMCQ dataset was generated and filtered using a single powerful LLM (Qwen3-235B), and the experiments are conducted using a model from the same family (Qwen2.5-Omni). This alignment could mean the model has an inherent advantage on this specific data distribution, potentially inflating the reported performance gains. I would be interesting to see the benefit of this training data and splits on a different base model.

**Questions:**

1. Regarding Section 2.4, you use Qwen3-235B to automatically evaluate samples. Did you consider using a hold-out evaluator from a completely different model family (e.g., GPT-4, Gemini, Claude) to validate this filtering process? Using the same model for generation and evaluation risks introducing a significant confirmation bias.
2. Your ACF method creates a binary weak/strong partition. Audio contribution, however, likely exists on a spectrum. Could you comment on the limitations of this hard binary split? Have you considered an alternative approach, such as a "soft" assignment where samples are weighted by their predicted audio-contribution strength during training?
3. You provide CoT annotations to "support CoT research" but do not use them for training. Do you have a hypothesis on how CoT fine-tuning would interact with your weak-to-strong paradigm? For instance, would you expect CoT to be more beneficial when applied to the 'strong' audio-contribution data, where complex, multi-step reasoning about audio events is presumably required?
4. The results show that the optimal SFT data strategy (Weak-AC vs. Mixed-AC) depends on the audio-contribution characteristics of the downstream benchmark. This suggests the primary benefit might be a distribution matching effect (i.e., aligning the SFT data distribution with the test distribution). How would you distinguish this effect from the more fundamental claim that your paradigm teaches the model a generalizable, robust reasoning process? Is it more about domain adaptation than creating a more capable model overall?

---

> ### Author Response · Authors · 2025-11-19
> **Response to Reviewer YhaS (Response to Weakness 1 & Question 3: CoT Experiments + Weakness 2 & Question 2: ACF Robustness and Soft Assignment)**
>
> We express our **sincere gratitude** for your thorough review and the "Excellent" rating regarding our contribution. Below, we address your concerns point-by-point. **All modified content has been marked in blue in the revised PDF.**
>
> ### **1. Response to Weakness 1 & Question 3: CoT Experiments**
>
> **Reviewer Comment:** Concerns regarding the lack of experiments with CoT data and how CoT fine-tuning interacts with the weak-to-strong paradigm.
>
> **Response:**
> We apologize for the lack of clarity regarding our CoT investigation. In this study, we focus more on the model's perception capability (which is more relevant to audio contribution) than its reasoning capability. Therefore, we do not incorporate CoT. However, we are indeed actively conducting extensive experiments related to CoT. These experiments will form the core focus of our **next major publication**. Please allow us to offer a brief, preliminary insight:
>
> We demonstrate that CoT improves reasoning in audio understanding. By integrating 'strong audio-contribution' data and other methods, we aim to **synergize this reasoning capability with robust fundamental perception.**
>
> **Action 1:** To **preserve the innovative edge** of our forthcoming publication, we respectfully request that the aforementioned details of CoT experiments remain **excluded from the current article**.
>
> **Action 2:** We have updated the first paragraph of **Section 4 Experiments** to include an announcement: *"Given our focus on perceptual fidelity rather than reasoning, CoT is excluded from the following evaluations."*
>
> ### **2. Response to Weakness 2 & Question 2: ACF Robustness and Soft Assignment**
>
> **Reviewer Comment:** Concerns about the justification/robustness of the three-model vote for ACF, and the potential for "soft" assignment.
>
> **Response:**
>
> **Regarding Model Selection for ACF:** We selected the voting models (R1-AQA, Audio-Flamingo2, Kimi-Audio) based on the state-of-the-art landscape available during our data construction phase (April 2025). To minimize bias and ensure voting fairness, we deliberately selected models from **distinct families where possible**, avoiding overlap with other available models like Qwen-Audio, Qwen2-Audio, and Audio-flamingo. Furthermore, **the significant improvement in perception metrics** achieved by models trained on our partitioned data (refer to **Appendix E.3, Table 8**) **serves as empirical validation that our current split is effective.**
>
> **Action:** We have updated the **Conclusion** to include these points: *"Future work could focus on establishing a more robust audio-contribution-based data partitioning for AudioMCQ, extending Audio-Contribution Filtering to general audio question-answering tasks, and further investigating CoT reasoning capabilities in audio understanding contexts."*
>
> **Regarding Soft Assignment:** We **agree entirely** with your insightful observation. A spectrum-based "soft" assignment mechanism represents a highly promising and critical direction for future research. This topic also constitutes a central component of **another article we are preparing** for imminent publication. We are very pleased to note the **strong consensus** between your perspective and our own findings on this matter.
>
> As with Weakness 1, we are happy to offer a brief, preliminary introduction to this specific research focus:
>
> Formally, let $\mathbf{a} = \{a_1, \dots, a_M\}$ denote the sequence of audio tokens and $\mathbf{q} = \{q_1, \dots, q_N\}$ represent the question text tokens. In the standard multimodal setting, the model predicts the target response tokens $\mathbf{y} = \{y_1, \dots, y_T\}$ conditioned on the concatenation of both modalities, $[\mathbf{a}; \mathbf{q}]$.
>
> To assess the dependency on textual context alone, we isolate the question tokens and calculate the likelihood of generating the ground-truth response $\mathbf{y}$ given only $\mathbf{q}$. We compute the **Perplexity (PPL)** of the response under this text-only condition, normalized by the sequence length $T$:
>
> $$
> \text{PPL}(\mathbf{y} | \mathbf{q}) = \exp\left( -\frac{1}{T} \sum_{t=1}^{T} \log P_\theta(y_t \mid \mathbf{q}, y_{<t}) \right)
> $$
>
> where $P_\theta$ represents the probability distribution parameterized by the model, and $y_{<t}$ denotes the preceding tokens in the response sequence.
>
> The **Weak Audio-Contribution Score**, denoted as $\mathcal{S}_{\text{weak}}$, can be defined as the inverse of this perplexity:
>
> $$
> \mathcal{S}_{\text{weak}} = \frac{1}{\text{PPL}(\mathbf{y} | \mathbf{q})}
> $$
>
> Intuitively, a lower perplexity (indicating higher predictability from text alone) results in a higher $\mathcal{S}_{\text{weak}}$, signifying that the model can infer the correct response with minimal reliance on the audio modality.
>
>
> **Action:** To safeguard the novelty of our upcoming research, we respectfully request that these prospective research details **remain excluded** from the current manuscript.

---

> > ### Author Response · Authors · 2025-11-19
> > **Response to Reviewer YhaS (Response to Weakness 3 & Question 1: Model Family Bias)**
> >
> > ### **3. Response to Weakness 3 & Question 1: Model Family Bias**
> >
> > **Reviewer Comment:** Concerns about potential performance inflation due to aligning the model family (Qwen) used for generation/filtering and experimentation, and the suggestion to use a hold-out model for validation.
> >
> > **Response:**
> > We acknowledge the concern regarding model alignment and potential performance inflation.
> >
> > **Regarding the Use of Non-Qwen LALMs:** It is currently **challenging** to find high-performing, open-source Large Audio Language Models (LALMs) that are entirely unrelated to the Qwen family and suitable for our post-training paradigm (GRPO).
> >
> > * Many strong LALMs, such as **Kimi-Audio** (https://github.com/MoonshotAI/Kimi-Audio) and **Midasheng** (https://github.com/xiaomi-research/dasheng-lm), utilize Qwen as their underlying Language Model component.
> > * Other models, like **Step-Audio2** (https://arxiv.org/abs/2507.16632), have already undergone advanced reinforcement learning (like GRPO) to the extent that **further fine-tuning becomes technically difficult (due to entropy collapse)**. The current ecosystem's structure makes it practically necessary to use a Qwen-family model for our experiments.
> >
> > **Regarding Quality Check with Different Models:** We appreciate the suggestion to use a hold-out evaluator (e.g., GPT-4 or Gemini). We initially used Qwen3-235B for automated quality filtering because **local inference** was necessary to **process the massive dataset efficiently**, guaranteeing timely data processing. We plan to explore calling external APIs (like ChatGPT's) for secondary, independent quality validation of our dataset in future updates.
> >
> > Under the limited conditions, to mitigate bias in the constructed data, we employed **rigorous quality control measures**, including multiple stages of algorithmic filtering and careful manual inspection during the dataset construction.
> >
> > **Action:** We have added **Appendix C: Quality Verification within the Pipeline**, which details our multi-stage quality control process (including manual checks) to demonstrate the reliability of the data.

---

> > > ### Author Response · Authors · 2025-11-19
> > > **Response to Reviewer YhaS (Response to Response to Question 4: Distribution Matching vs. Robust Reasoning)**
> > >
> > > ### **4. Response to Question 4: Distribution Matching vs. Robust Reasoning**
> > >
> > > **Reviewer Comment:** Is the benefit primarily due to distribution matching (domain adaptation) rather than learning a generalizable reasoning process?
> > >
> > > **Response:**
> > > We respectfully **maintain that our paradigm instills generalizable robustness,** distinct from simple domain adaptation.
> > >
> > > 1.  **Universal Perception Gain:** Regardless of the SFT strategy (Mixed vs. Weak), **both paradigms outperform the baseline (Mixed-to-Mixed).** This is driven by the **GRPO stage using strong audio-contribution data**, which significantly enhances the model's fundamental "listening" capability. This gain is consistent across **all benchmarks**, indicating a **generalizable improvement** in processing audio signals (see **Figure 5**).
> > > 2.  **Role of SFT:** We view the SFT data selection as a downstream **engineering choice** to align the model with specific task formats **under the premise** the fundamental perception capabilities **have been secured by GRPO**. This alignment **does not negate** the general robustness gained during the reinforcement learning phase.
> > >
> > > **Action:** We have clarified this distinction in the final paragraph of **Section 4.2**:
> > > *"In contrast, GRPO with strong audio-contribution data yields significant improvements across all benchmarks and their AC-strong splits, reflecting a substantial gain in the model's perception capability, particularly pronounced on the strong audio-contribution benchmarks (MMAR and MMSU). Second, SFT data selection should align with specific downstream task characteristics."*
> > >
> > > Furthermore, we wish to express our viewpoint: We believe the limitation is not that our post-training method lacks generality, but rather that **the design of current models and most benchmarks has yet to sufficiently address the issue of audio contribution.** Consequently, the audio contributions in existing benchmarks vary significantly.
> > >
> > > We firmly believe that a Large Audio Language Model **should derive its answer entirely based on the audio it hears**, rather than guessing based on its own prior knowledge and reasoning capacity; otherwise, this causes significant confusion for the user. Similarly, an LALM benchmark **should inherently possess a strong audio contribution.**
> > >
> > > We posit that our designed strong audio-contribution GRPO is more critical than the SFT stage because it **guides the model in the correct direction.** SFT, as we previously mentioned, is merely an **engineering choice** necessitated by the current context.
> > >
> > > ***
> > > **Additional Update: Refined Checkpoint Selection for Mixed-to-Strong Strategy**
> > >
> > > In the process of finalizing our experimental records, we conducted a comprehensive retrospective review of the training trajectory for the **Mixed-to-Strong** paradigm. We identified an earlier checkpoint—specifically at **500 GRPO steps** (following the initial 1000 mixed-SFT steps)—that was previously overlooked.
> > >
> > > Upon closer inspection, this **500-step checkpoint** not only matches the originally selected **900-step checkpoint** on the validation set (MMAU-test-mini-4k) but actually yields **superior generalization** on the primary benchmarks. We have adopted this checkpoint as the new optimal model for the Mixed-to-Strong paradigm.
> > >
> > > The updated performance comparison is presented below:
> > >
> > > | Benchmark | **New 500-step Checkpoint** | Previous 900-step Checkpoint |
> > > | :--- | :---: | :---: |
> > > | **MMAU-test-mini** | $\mathbf{76.4}$ | $75.8$ |
> > > | **MMAU** | $\mathbf{75.1}$ | $74.6$ |
> > > | **MMAR** | $67.0$ | $\mathbf{67.1}$ |
> > > | **MMSU** | $\mathbf{71.7}$ | $70.7$ |
> > >
> > > **Action:** We have updated **Table 2** and the corresponding analysis in the manuscript to reflect the performance of this improved 500-step model. We have added a **Figure 4** showing the selection of the best checkpoint during the experiments.
> > > ***
> > >
> > > Thank you again sincerely for your interest in our paper.

---

> ### Comment · Reviewer_YhaS · 2025-11-26
>
> The authors have clarified the scope of the work. Although some of the points raised are left for future work, the practical contributions of this paper are still valuable. I will maintain my positive rating of 6.

---

> > ### Author Response · Authors · 2025-11-26
> >
> > Thank you for your positive evaluation and constructive feedback on our work. We **greatly appreciate** your recognition of the contributions of our paper :)

---

### Official Review · Reviewer_xxSA · 2025-11-01

**Soundness:** 2
**Presentation:** 3
**Contribution:** 2
**Rating:** 4
**Confidence:** 4

**Summary:**

The paper studies the post-training strategies for training large audio language models. The paper explores multistage post-training strategies such as RL followed by SFT, instead of single-stage post-training. The paper uses the multi-stage post-training to improve the zero audio-contribution phenomenon in LALMs, where models derive correct answers solely from textual information without processing audio content. The paper proposes a comprehensive MCQ dataset to facilitate large-scale post-training.

**Strengths:**

The paper is well-written and explores important yet underexplored multistage post-training strategies. The proposed dataset is comprehensive and focuses on designing MCQ that require the LALM to pay attention to audio to answer correctly. The paper achieves strong performance across several benchmarks, such as MMAU and MMAR.

**Weaknesses:**

The synthetic data generation, followed by post training, has been used before to improve performance of prior models such as R1-Qwen and R1-Omni.

The performance improvement across the two benchmarks is very small; for example, on the MMAU test and test-mini benchmark, the performance is just 0.2, and on MMAR is <2 (Table 5), despite using significantly more data for post-training than the existing methods, such as Omni-R1.

**Questions:**

The experiments for the two proposed post-training paradigms, Weak-to-Strong and Mixed-to-Strong, are poorly designed. Why is the number of training steps across the three training experiments different?

Line 442: “Mixed-to-Mixed approach (1000 steps SFT + 400 steps GRPO with mixed audio contribution data)”
“Weak-to-Strong paradigm (750 steps weak audio-contribution SFT + 100 steps strong audio-contribution GRPO)”
“Mixed-to-Strong approach (1000 steps mixed audio-contribution SFT + 900 steps strong audio-contribution GRPO)”
This makes drawing conclusions about the post-training strategies difficult. Is the “Mixed-to-Strong” strategy better than the “Mixed-to-Mixed” strategy because of how data is used, or is it simply due to the larger number of training steps used for the “Mixed-to-Strong” strategy?

---

> ### Author Response · Authors · 2025-11-17
> **Response to Reviewer xxSA (Addressing Weakness 1: Synthetic Data Generation and Prior Work)**
>
> We sincerely appreciate your thorough review and constructive feedback on Submission 10375. We address the weakness 1 you raised point-by-point below. **All modified content has been marked in blue in the PDF.**
>
> ---
>
> **1. Synthetic Data Generation and Prior Work (R1-Qwen and R1-Omni)**
>
> **Reviewer Comment:** "The synthetic data generation, followed by post training, has been used before to improve performance of prior models such as R1-Qwen and R1-Omni."
>
> **Response:**
>
> We acknowledge that the concept of using synthetic data for post-training has been explored in prior work like R1-Qwen and Omni-R1. However, we respectfully assert that our data generation pipeline differs from R1-Qwen (R1-AQA) and Omni-R1 in the following key aspects:
>
> i. **More Careful Data Source Selection:** We prioritized data sources, ensuring $4/7$ are **fully human-annotated** and $3/7$ utilize **a rigorous pipeline with subsequent human auditing** (refer to Table 1). These sources are dedicated to **pure audio understanding**. In contrast, R1-Qwen used the AVQA dataset (Audio-Visual Question Answering) and Omni-R1 utilized VGGSound, both often requiring less focused audio processing.
>
> ii. **Multi-Step Generation Pipeline with LLM:** Instead of a single-step synthesis, our pipeline involves **six distinct steps** (see Figure 1), with **four steps utilizing Qwen3-235B**. Crucially, we designed **distinct prompts for specific tasks** like Question Type Generation and Choices Generation for different content categories (sound/music/speech and temporal aspects) (see Appendix B.1 and B.2). This complex, iterative process significantly enhances data quality and **enriches the data with additional information** (e.g., question type, two types of Chain-of-Thought (CoT)). R1-AQA simply replaced "video" with "audio" in the AVQA dataset, and Omni-R1 used a comparatively simpler, single-step prompt.
>
> iii. **Novel Audio Contribution Filtering:** We innovatively integrated an **Audio Contribution Filtering** technique into the data construction process, resulting in **Weak Audio Contribution** and **Strong Audio Contribution** subsets (see Section 3.2). This novel methodology provides a new standard for constructing datasets and benchmarks aimed at LALMs.
>
> iv. **Rigorous Quality Verification and Pipeline Optimization:** We implemented stringent human and automated quality checks at every step of the pipeline (Figure 1). Each generated data point undergoes Automated Validation for format and structural integrity, with necessary regeneration if issues are found. Furthermore, $2-3$ human annotators randomly sampled and audited the generated data. Any detected issues led to modifications of the Automated Validation program or the LLM prompts, followed by the regeneration of all data through the updated pipeline. We apologize for not explicitly detailing this process initially. **We have now added a new section**, **Appendix C: Quality Verification within the Pipeline**, for reference.
>
> ---

---

> ### Author Response · Authors · 2025-11-17
> **Response to Reviewer xxSA (Addressing Weakness 2: Performance Improvement Magnitude)**
>
> We appreciate the opportunity to clarify and strengthen our submission based on your insightful feedback. We address the weakness 2 point-by-point below. **All modified content has been marked in blue in the PDF.**
>
> ---
>
> **2. Performance Improvement Magnitude**
>
> **Reviewer Comment:** "Despite using significantly more data for post-training than the existing methods, such as Omni-R1, the performance improvement across the two benchmarks is very small; for example, on the MMAU test and test-mini benchmark, the performance is just 0.2, and on MMAR is <2 (Table 5)."
>
> **Response:**
>
> We appreciate the reviewer's attention to the magnitude of performance improvements.
> We would like to **reiterate the following two points**:
>
> i. **The Improvement in Perceptual Capability is Actually Evident:**
>
> While the gains on the MMAU benchmark are modest, the improvement on **MMAR** is notably significant ($\mathbf{+1.9\%}$ over Omni-R1 and $\mathbf{+2.3\%}$ over All Data GRPO). We attribute this disparity to the nature of the benchmarks:
>
> * **MMAR** is a benchmark with a strong requirement for detailed audio perception.
> * **MMAU** is characterized as a **weak audio-contribution** dataset, where up to $\mathbf{58.4\%}$ of the questions can be correctly answered based on textual context alone (refer to Table 2).
>
> The substantial $\mathbf{1.9\%}$ & $\mathbf{2.3\%}$ gain on MMAR confirms that our multi-stage training paradigm significantly enhances the model's **"ability to perceive and process audio"** (as visually supported by Figure 5). We posit that the marginal improvement on MMAU, despite this enhanced capability, is a logical consequence of **MMAU's inherent weak audio-contribution nature**.
>
> ii. **Motivation for Integrating a lot more SFT Data into the Multi-Stage Paradigm:**
>
>  We emphasize that the core motivation for Integrating SFT Data is **to address the "data oversupply"** scenario common in LALM post-training. In this scenario, the RL (GRPO) stage often **converges very quickly, limiting the amount of overall data that can be effectively utilized**. Traditional SFT-RL frameworks may fail if data is randomly allocated to the SFT stage. Our paradigm is designed to efficiently utilize a much larger volume of data in the SFT stage. We believe that with **the trend of scaling of post-training**, the performance gap between our STF-RL method and pure RL **will widen**.
>
> **Overall**, to provide the context above within the manuscript, we have added the following clarifications:
>
> * **Section 4.2 (Last Paragraph, about the gain in perception capability):** "In contrast, GRPO with strong audio-contribution data yields significant improvements across all benchmarks and their AC-strong splits, reflecting a substantial gain in the model's perception capability, particularly pronounced on the strong audio-contribution benchmarks (MMAR and MMSU)."
> * **Introduction (Third Paragraph, about the weakness of Mixed-to-Mixed):** "Despite utilizing more data, two-stage paradigms (e.g., SFT followed by RL) may not consistently outperform single-stage post-training, which imposes an effective upper bound on the data volume for post-training."
>
> We believe these points clarify the strategic value and differentiated impact of our proposed multi-stage post-training paradigm.
>
> ---

---

> ### Author Response · Authors · 2025-11-17
> **Response to Reviewer xxSA (Answering Question 1: Experimental Design of Post-training Paradigms)**
>
> We are deeply grateful for your comprehensive review and insightful feedback on Submission 10375. We address the question you raised point-by-point below. **All modified content has been marked in blue in the PDF.**
>
> ---
>
> **1. Experimental Design of Post-training Paradigms**
>
> **Reviewer Comment:** "The experiments for the two proposed post-training paradigms, Weak-to-Strong and Mixed-to-Strong, are poorly designed. Why is the number of training steps across the three training experiments different? ... This makes drawing conclusions about the post-training strategies difficult. Is the 'Mixed-to-Strong' strategy better than the 'Mixed-to-Mixed' strategy because of how data is used, or is it simply due to the larger number of training steps used for the 'Mixed-to-Strong' strategy?"
>
> **Response:**
>
> We apologize for the confusion regarding the training steps and the difficulty in drawing conclusions.
>
> We would like to clarify that we **maintained a consistent training step limit for each stage** across the experiments, and the final reported step counts were selected based on the optimal checkpoint chosen using the **MMAU-test-mini-4k** validation set (refer to Appendix E.2).
>
> To provide visual clarity and directly address the concern about training step disparity, we have added **Figure 4: Performance comparison of three training approaches on MMAU-test-mini-4k for optimal checkpoint selection**.
>
> This figure clearly shows two key points:
>
> 1.  In the left panel of Figure 4, the **Weak-to-Strong** model reaches its optimal performance very quickly (after only **100 RL steps**). Its total data usage is thus significantly lower than both Mixed-to-Mixed and Mixed-to-Strong, ruling out "more training steps" as the cause of superiority.
> 2.  In the right panel of Figure 4, starting from the same SFT checkpoint, the **Mixed-to-Strong** strategy **consistently and significantly outperforms** the **Mixed-to-Mixed** strategy throughout the RL phase. This confirms that the superior performance is **due to the way the data is used** (i.e., the paradigm itself), not merely the number of steps.
>
> Therefore, it can be seen that the superior performance of both of our two paradigms is not due to a greater number of training steps.
>
> **Crucial Update and Correction:**
>
> **Following your question**, we re-evaluated all checkpoints for the **Mixed-to-Strong** paradigm. We discovered that the checkpoint after **"1000 steps mixed audio-contribution SFT + 500 steps strong audio-contribution GRPO"** was missed. This **$500$-step** checkpoint exhibited performance **equal to** the previously selected **$900$-step** checkpoint on MMAU-test-mini-4k.
>
> To definitively exclude the hypothesis that a larger number of training steps leads to better results, we have opted to use the $\mathbf{500\text{-step GRPO checkpoint}}$ as our new optimal checkpoint for the Mixed-to-Strong paradigm.
>
> The new performance metrics for the $\mathbf{500\text{-step checkpoint}}$ are:
> | Benchmark | New 500-step Checkpoint Performance | Previous 900-step Checkpoint Performance |
> | :---: | :---: | :---: |
> | MMAU-test-mini | $\mathbf{76.4}$ | $75.8$ |
> | MMAU | $\mathbf{75.1}$ | $74.6$ |
> | MMAR | $67.0$ | $\mathbf{67.1}$ |
> | MMSU | $\mathbf{71.7}$ | $70.7$ |
>
> This result, showing superior performance on key benchmarks (MMAU and MMSU) with fewer training steps, provides even stronger evidence for our conclusion that the **Mixed-to-Strong** paradigm is inherently more effective due to its structured use of data.
>
> We would like to **express our sincere gratitude for this comment.** Your insightful query prompted us to re-evaluate our checkpoints, which led us to identify this superior model. This finding **significantly strengthens the central arguments of our paper** and has tangibly improved the overall quality of our manuscript.
>
> We have updated the manuscript accordingly, specifically in **Table 5**, **Table 8**, **Figure 5**, the **Abstract**, **Introduction (third paragraph end)**, **Section 4.2 (end)**, and the **Conclusion**, reflecting the new, stronger results from the 500-step checkpoint.
>
> As a final note, we are actively working to **release the AudioMCQ** dataset, as well as the checkpoints for the **Weak-to-Strong** and **Mixed-to-Strong** models, very soon.
>
> Thank you once again for your rigorous review and invaluable suggestions.

---

> > ### Comment · Reviewer_xxSA · 2025-11-27
> >
> > The authors have answered most of my concerns and I have increased my score.

---

> > > ### Author Response · Authors · 2025-11-29
> > >
> > > Thank you for your professional and constructive feedback. Although we are unable to see the updated score due to recent changes in the conference's review policy, we are **deeply grateful for your recognition of our responses** and your decision to raise the score. We **truly appreciate** your time and effort in reviewing our work.

---

### Meta-Review · Area_Chair_Feig · 2026-01-08

**Summary:**

Reviewers flagged limited novelty relative to prior pipelines, small gains on some benchmarks and potentially confounded paradigm comparisons due to unequal step counts/checkpointing. Another concern was lack of open-ended QA and ack of CoT training/ablation , and also aspects of robustness and possible bias since generation and training are in the same type of models.

**Reviewer Concerns:**

The “confounded comparison / unequal steps” concern was addressed and the novelty concern was articulated by pointing to concrete pipeline differentiators. The authors also added a quality-verification appendix. Small gains concern was addressed by pointing to stronger improvements on audio-critical benchmarks. The MCQ-only limitation was not resolved. Many concerns seem to remain, such as no experiments using CoT or isolating its effect, limited ablation/robustness analysis and sensitivity to evaluator/model choice.

**Reviewer Scores:**

Ignoring the disputed review, the rebuttal would have bumped up one borderline reviewer by resolving concerns about novelty, benchmark interpretation, and experimental confounds. Two reviewers were already positive and the rebuttal did not introduce new weaknesses.

---

### Decision · Program_Chairs · 2026-01-26

Accept (Poster)